# Mesenchyme instructs growth while epithelium directs branching in the mouse mammary gland

Qiang Lan[1], Ewelina Trela[1], Riitta Lindström[1], Jyoti Prabha Satta[1], Beata Kaczyńska[1], Mona M Christensen[1], Martin Holzenberger[2], Jukka Jernvall[1,3], Marja L Mikkola[1]*

[1]Cell and Tissue Dynamics Research Program, Institute of Biotechnology, Helsinki Institute of Life Science (HiLIFE), University of Helsinki, Helsinki, Finland; [2]Sorbonne University, INSERM, Research Center Saint-Antoine, Paris, France; [3]Department of Geosciences and Geography, University of Helsinki, Helsinki, Finland

**Abstract** The mammary gland is a unique organ that undergoes dynamic alterations throughout a female's reproductive life, making it an ideal model for developmental, stem cell and cancer biology research. Mammary gland development begins in utero and proceeds via a quiescent bud stage before the initial outgrowth and subsequent branching morphogenesis. How mammary epithelial cells transit from quiescence to an actively proliferating and branching tissue during embryogenesis and, importantly, how the branch pattern is determined remain largely unknown. Here, we provide evidence indicating that epithelial cell proliferation and onset of branching are independent processes, yet partially coordinated by the Eda signaling pathway. Through hetero-typic and heterochronic epithelial-mesenchymal recombination experiments between mouse mammary and salivary gland tissues and ex vivo live imaging, we demonstrate that unlike previously concluded, the mode of branching is an intrinsic property of the mammary epithelium whereas the pace of growth and the density of ductal tree are determined by the mesenchyme. Transcriptomic profiling and ex vivo and in vivo functional studies in mice disclose that mesenchymal Wnt/ß-catenin signaling, and in particular IGF-1 downstream of it critically regulate mammary gland growth. These results underscore the general need to carefully deconstruct the different developmental processes producing branched organs.

*For correspondence:
marja.mikkola@helsinki.fi

**Competing interest:** The authors declare that no competing interests exist.

## Editor's evaluation

In this valuable study, the authors use classical embryonic tissue recombination and pharmacological manipulation of explants in conjunction with cutting edge 3D imaging of tissue derived from sophisticated reporter and knock-out mouse models, as well as transcriptomic analyses, to delineate and dissect regulatory pathways critical for early mammary development, specifically focusing on cell proliferation, and ductal branching. The conclusions are convincing and the findings will be of interest to the community of biologists interested in the cellular and molecular mechanisms of (early) mammary gland development, as well as to a broader community of developmental biologists studying branching morphogenesis in tissues such as lung, kidney and salivary gland.

## Introduction

Branching morphogenesis is a common developmental process driving the formation of a number of organs including lung, kidney, salivary, and mammary gland (*Lang et al., 2021*). Although some

fundamental principles are shared, each organ employs its unique branching strategy – mode and density of branching – to achieve the proper architecture tailored to its function (*Goodwin and Nelson, 2020*; *Lang et al., 2021*; *Myllymäki and Mikkola, 2019*). In recent decades, significant advancements have been made in unraveling the underlying mechanisms of branching morphogenesis in various organs and species. However, many questions remain unanswered, especially regarding the mammary gland as much of the research focus has been on its postnatal growth (*Goodwin and Nelson, 2020*; *Lang et al., 2021*). Yet, mammary gland morphogenesis commences already during fetal life by formation of placodes, local epithelial thickenings, in the flanks of the fetus. How these early steps of branching morphogenesis differ between mammary gland and other organs remains poorly understood.

In mice, five pairs of mammary placodes emerge around embryonic day 11 (E11). Placodes invaginate by E13 giving rise to buds that are now surrounded by condensed, mammary-specific mesenchyme (*Sakakura et al., 2013*; *Spina and Cowin, 2021*; *Watson and Khaled, 2020*). Mammary buds stay relatively non-proliferative until E15-E16 when they sprout toward the adjacent 'secondary' mammary mesenchyme, the fat pad precursor tissue that later gives rise to the adult stroma. Branching begins at E16, and by E18 (1–2 days prior to birth) mammary rudiments have developed into small ductal trees with 10–25 branches (*Lindström et al., 2022*; *Myllymäki and Mikkola, 2019*). In contrast to the postnatal bilayered mammary epithelium consisting of outer basal and inner luminal cells, embryonic mammary rudiments undergo branching as a solid mass of epithelial cells without lumen. Mammary rudiments initially consist of multipotent precursors that become restricted to basal and luminal lineages during later stages of embryogenesis (*Lilja et al., 2018*; *Wuidart et al., 2018*). The mechanisms governing the exit from quiescence and acquisition of branching ability are still enigmatic. During puberty, stochastic distribution of proliferating mammary stem cells drives the non-stereotypic branching of pubertal mammary gland (*Scheele et al., 2017*). However, whether a causal link exists between onset of proliferation and initial outgrowth in embryonic mammary gland development is currently unknown.

Reciprocal epithelial-mesenchymal tissue interactions are critical for mammary gland development at all stages. Many signaling pathways essential for mammary placode and bud formation have been identified, but the paracrine factors regulating branching during embryogenesis are less well understood (*Cowin and Wysolmerski, 2010*; *Hiremath and Wysolmerski, 2013*; *Spina and Cowin, 2021*; *Watson and Khaled, 2020*). The tumor necrosis factor family member ectodysplasin A1 (Eda) is one such mesenchymal factor: Eda deficiency compromises ductal growth and branching, while mice overexpressing Eda exhibit a dramatic ductal phenotype with precocious sprouting and excessive growth and branching (*Elo et al., 2017*; *Voutilainen et al., 2012*). In addition, the Wnt and fibroblast growth factor (Fgf) pathways are likely involved (*Cowin and Wysolmerski, 2010*; *Lindström et al., 2022*), but the early developmental arrest observed in mice where these pathways are inactivated (*Chu et al., 2004*; *Mailleux et al., 2002*) has hampered elucidation of their exact roles during branching morphogenesis.

Importantly, the current paradigm posits that the mesenchyme specifies the epithelial branching pattern in all branched organs (*Lang et al., 2021*; *Myllymäki and Mikkola, 2019*). This conclusion stems from tissue recombination experiments where epithelia and mesenchymes of different origins have been exchanged: lung mesenchyme instructs the kidney epithelium to adopt a lung-type branching pattern while organ-specific mode of branching is maintained in homotypic tissue recombinants (*Kispert et al., 1996*; *Lin et al., 2003*). The same conclusion was drawn from the pioneering experiments involving salivary gland mesenchyme and mammary gland epithelium. Even though the mammary epithelium retained its cellular identity, the branch pattern was reported to be salivary gland-like: branches formed at higher density and by tip clefting rather than lateral branching (*Kratochwil, 1969*; *Sakakura et al., 1976*). In addition, salivary gland mesenchyme promoted much faster growth. Although the evidence from the early experiments appears compelling, the underlying molecular basis remained elusive.

To uncover the regulation of mammary gland branching, we first revisited the heterochronic tissue recombination using mammary tissues. Our results show that the timing of the initial branching is epithelium-dependent, yet epithelial-mesenchymal interactions are indispensable for the outgrowth to occur. In strong contrast to the previous reports and to the paradigm of the role of the mesenchyme in directing branching (*Kratochwil, 1969*; *Sakakura et al., 1976*), live imaging disclosed that salivary

gland mesenchyme failed to switch the mode of mammary branching into salivary-like. This implies that branch pattern formation is an intrinsic property of the mammary epithelium. Nevertheless, salivary mesenchyme had a major growth-promoting effect on the mammary epithelium once it had acquired branching capacity. Transcriptomic profiling of mammary and salivary gland mesenchymes identified mesenchymal Wnt/ß-catenin pathway and its downstream target *Igf1* as potential drivers of epithelial growth, thereby deconstructing mode of branching from growth control in mammary development.

## Results

### The timing of onset of branching is an intrinsic property of the mammary epithelium

To assess whether timing of the mammary initial branching can be influenced by tissues of different developmental stages, we performed heterochronic epithelial-mesenchymal recombination experiments. To this end, we used tissues micro-dissected from fluorescently labeled transgenic mice allowing day-to-day imaging, as well as evaluation of the purity of the tissue compartments (*Figure 1A* and *Figure 1—figure supplement 1A and B*). Because anterior mammary glands are more advanced in their development than the posterior ones (*Lindström et al., 2022*), only mammary glands 1–3 were used throughout the study, unless otherwise specified, to avoid any biases caused by the asynchrony.

It has been previously shown that early (E12) mammary mesenchyme does not alter the onset of branching of the mammary epithelium (E12 to E16) in ex vivo tissue recombination experiments (*Kratochwil, 1969*). However, the ability of late mammary mesenchyme to advance epithelial outgrowth and branching has not been assessed. To answer this question, we recombined E13.5 mammary epithelia (bud stage) with E13.5, E15.5, or E16.5 (when the very first branches are evident) mammary mesenchymes. In the control explants (E13.5 epithelia with E13.5 mesenchyme), branching started after 3–4 days of culture (*Figure 1B and C*), in good agreement with development in vivo. No precocious branching was observed when 'older' mesenchyme was used: when E13.5 epithelia were cultured with either E15.5 or E16.5 mesenchyme, branching was again evident only after 3–4 days of culture (*Figure 1B and C*). As an additional control, we performed similar experiments as described by *Kratochwil, 1969*, and cultured E13.5, E15.5, or E16.5 mammary epithelia with E13.5 mammary mesenchyme (*Figure 1D*). As previously reported, all epithelia branched in E13.5 mesenchyme, and outgrowth started after 3–4, 1–2, and 0–1 days of culture, respectively (*Figure 1E*), correlating with the stage of epithelium and its developmental pace in vivo.

Next, we asked whether the mesenchyme is needed for initiation of branching. To this end, we utilized a mesenchyme-free 3D mammary organoid technique to culture micro-dissected intact mammary rudiments in a serum-free medium with growth supplements (*Lan et al., 2022*; *Figure 1F*). In the 3D Matrigel matrix, E16.5 mammary epithelia generated large branching trees in just 3 days (*Figure 1G and H*, and *Figure 1—figure supplement 1C*), whereas epithelia from earlier stages (E13.5 to E15.5) consistently failed to branch even after 8 days of culture. Some specimens enlarged in size, yet they failed to progress, except for occasional E15.5 epithelia that generated a few branches (*Figure 1G–I* and *Figure 1—figure supplement 1C*). To assess if apoptosis could explain the failure of E13.5-E15.5 epithelia to generate outgrowths, we quantified cleaved caspase-3$^+$ cells after 2 days in 3D culture. A significant increase in apoptosis was observed in E14.5 epithelia compared to E16.5 epithelia (*Figure 1—figure supplement 1D and E*). However, ~40% of the E14.5 samples exhibited low levels of apoptosis, similar to that observed in E16.5 samples, suggesting that apoptosis may contribute to, but is unlikely to be the primary factor limiting the branching capacity of E13.5-E15.5 mammary epithelia in mesenchyme-free 3D culture.

Besides confirming previous observations (*Kratochwil, 1969*), our results reveal that mesenchymes from advanced embryonic developmental stages could not alter the pace of epithelial outgrowth, yet epithelial-mesenchymal interactions are indispensable for the mammary epithelium to acquire branching ability.

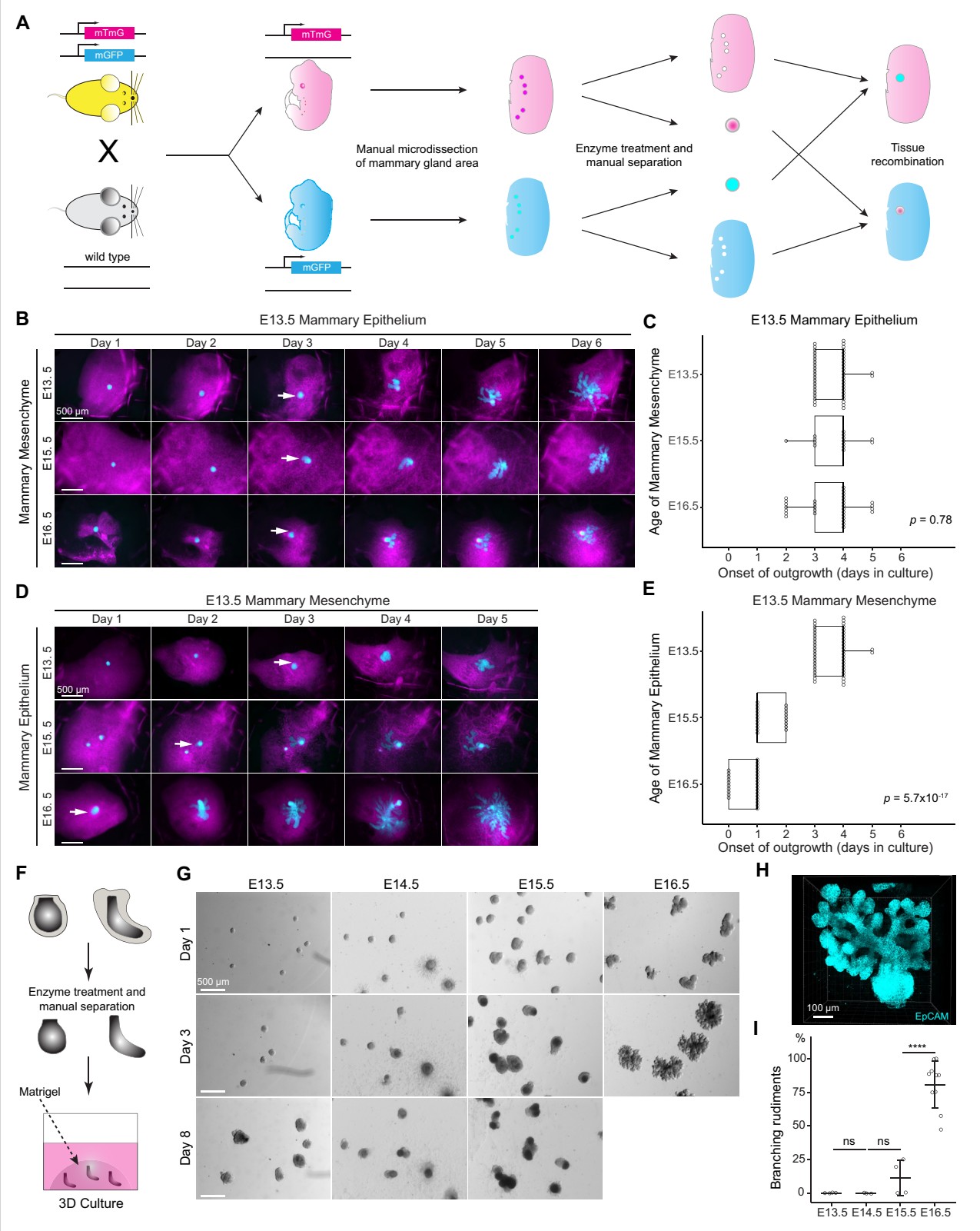

**Figure 1.** The timing of mammary gland outgrowth is an inherent property of the epithelium. (**A**) A scheme illustrating the experimental procedure used in tissue recombination experiments. (**B**) Representative images showing the onset of outgrowth of E13.5 mammary epithelia recombined with E13.5, E15.5, or E16.5 mammary mesenchymes, respectively. The appearance of the primary outgrowth is indicated with arrow. Scale bar, 500 µm. (**C**) Quantification of the time (in days) required for onset of the branching. Data were pooled from three to six independent experiments of E13.5 mammary

*Figure 1 continued on next page*

*Figure 1 continued*

epithelia recombined with E13.5 (n=46 explants), E15.5 (n=14), and E16.5 (n=30) mammary mesenchymes. Statistical significance was assessed with the Kruskal–Wallis test. (**D**) Representative images showing onset of outgrowth of E13.5, E15.5, and E16.5 mammary epithelia recombined with E13.5 mammary mesenchymes. The appearance of the primary outgrowth is indicated with arrows. Scale bar, 500 µm. (**E**) Quantification of the time (in days) required for the onset of the primary outgrowth. Data were pooled from three to six independent experiments of E13.5 (n=46 explants), E15.5 (n=20) and E16.5 (n=27) mammary epithelia recombined with E13.5 mammary mesenchyme. Statistical significance was assessed with the Kruskal–Wallis test. (**F**) A scheme illustrating the 3D culture of intact, mesenchyme-free epithelial mammary rudiments. (**G**) Representative images showing the growth of E13.5, E14.5, E15.5, and E16.5 epithelial mammary rudiments in 3D culture; only E16.5 mammary rudiments were capable of branching (see also *Figure 1—figure supplement 1C*). Scale bar, 500 µm. (**H**) Representative 3D projection image of an EpCAM-stained E16.5 mammary rudiment after three days of 3D culture in Matrigel. Scale bar, 100 µm. (**I**) Quantification of branching mammary rudiments in 3D culture. Data are presented as percentage of branching mammary rudiments (mean ± SD) from a total of 4 (E13.5), 3 (E14.5), 4 (E15.5), and 10 (E16.5) independent experiments (each with minimum 6 rudiments in culture). The statistical significances were assessed using unpaired two-tailed Student's *t*-test with Bonferroni correction. ns, non-significant; ****, p<0.001.

The online version of this article includes the following source data and figure supplement(s) for figure 1:

**Source data 1.** Source data of quantifications represented as graphs in *Figure 1C, E1*.

**Figure supplement 1.** Mesenchyme does not alter the timing of mammary gland outgrowth but is required for initiation of branching.

**Figure supplement 1—source data 1.** Source data of quantifications represented as graphs in *Figure 1—figure supplement 1E*.

## Basal-cell biased proliferation is activated in mammary epithelium prior to initiation of branching

Next, we sought to determine which mammary epithelial properties are required for the onset of branching. The majority of mammary epithelial cells are quiescent at the placode and bud stages (*Balinsky, 1950*; *Lee et al., 2011*; *Trela et al., 2021*), and proliferation is thought to resume when branching begins at around E16 (*Balinsky, 1950*). Such coincidence suggests that activation of proliferation may closely cooperate with, or even drive the onset of branching. To gain more insight into the quiescent stage of the embryonic mammary primordium, we first quantified the volume of the mammary epithelium with the aid of 3D surface renderings of EpCAM-stained specimens (*Figure 2A*). The volume of mammary rudiments steadily increased from E13.5 to E16.5 (*Figure 2B*), whereas quantification of the branch (tip) number showed that active branching did not take place until E16.5 (*Figure 2C*).

To analyze epithelial proliferation between E13.5 and E16.5, we investigated cell cycle dynamics using the Fucci2a mouse model derived from the *Rosa26*^Fucci2a flox/Fucci2a flox^ mice (*Mort et al., 2014*) by permanently deleting the stop cassette. This resulted in mice where cells in S/G2/M phase of the cell cycle constitutively express nuclear mVenus while cells in G1/G0 express nuclear mCherry. The ratios of mammary epithelial cells in S/G2/M and G1/G0 phases were quantified in 3D after whole-mount staining with EpCAM (*Figure 2D*). In line with the previous report (*Trela et al., 2021*), only ~20% of mammary epithelial cells were in S/G2/M phase at E13.5, with no apparent change at E14.5 (*Figure 2E*). However, the proportion of S/G2/M cells significantly increased at E15.5 but plateaued and even slightly decreased at E16.5 when branching was evident (*Figure 2E*). Notably, the proliferating cells exhibited a tendency to localize close to the epithelial-mesenchymal interface (basal layer) starting from E15.5 (*Figure 2D*).

Next, we examined in more detail whether the proliferative cells display any bias in their distribution at E13.5-E16.5. Due to the absence of clear spatial segregation of basal and luminal lineage markers during these early developmental stages (*Wuidart et al., 2018*), we focused on the location of the cells and measured the distance of each nucleus to the surface of epithelial mammary rudiments (i.e. epithelial-mesenchymal border) in 3D (*Figure 2F*). Distribution of all nuclei revealed a significant fraction of cells localizing within 10 µm distance from the epithelial surface (dashed line in *Figure 2G*), corresponding well with the confocal images showing radially organized, basally-located elongated cells in the same position (*Figure 2D and F*). Next, we stratified the epithelial cells to basal (nuclear distance less than or equal to 10 µm from the surface) and inner ("luminal") (nuclear distance more than 10 µm) ones and quantified the ratios of S/G2/M and G1/G0 cells in each compartment (*Figure 2H*). At E13.5 and E14.5, the proportion of S/G2/M cells was higher in the inner compartment, though the difference was statistically significant only at E14.5. However, concomitant with the overall increase in proliferation (*Figure 2E*), there was a switch in the proportion of S/G2/M and G1/G0 cells at E15.5 and E16.5, basal cells being significantly more proliferative.

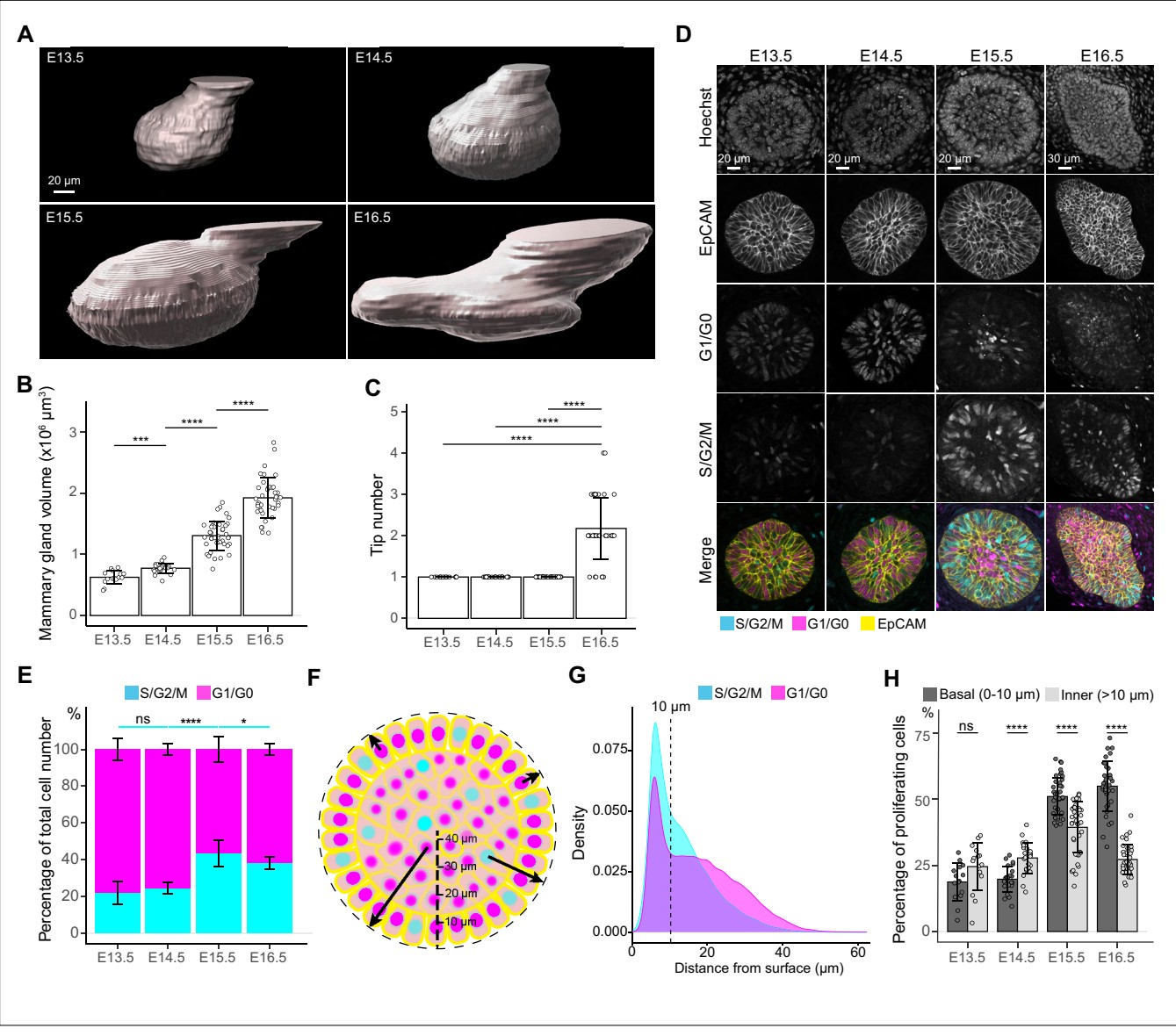

**Figure 2.** Cell cycle dynamics in embryonic mammary glands. (**A**) Representative 3D surface rendering images of EpCAM-stained E13.5, E14.5, E15.5, and E16.5 epithelial mammary rudiments, based on 3D confocal imaging. Mammary gland 2 is shown. Scale bar, 20 μm. (**B, C**) Quantification of epithelial mammary gland volume (**B**) and number of branching tips (**C**), $n_{E13.5}$=15, $n_{E14.5}$=24, $n_{E15.5}$=41, $n_{E16.5}$=36. (**D**) Confocal optical sections of whole mount-stained mammary glands from E13.5, E14.5, E15.5, and E16.5 Fucci2a embryos stained with EpCAM. Scale bars, 20 μm (E13.5-E15.5) and 30 μm (E16.5). (**E**) Quantification of the proportions of all epithelial cells in S/G2/M and G1/G0 phases. Altogether, 15 glands (in total 9228 cells) from three E13.5 embryos, 24 glands (in total 17,599 cells) from five E14.5 embryos, 41 glands (in total 40,431 cells) from eight E15.5 embryos, and 36 glands (in total 50,574 cells) from seven E16.5 embryos were analyzed. (**F**) A schematic image illustrating how the distance of cells (center of the nucleus) was quantified with respect to the surface of mammary rudiments. (**G**) Density plot showing the distribution of the distance of nuclei in S/G2/M and G1/G0 phase to the surface of the mammary rudiment. Density plot revealed that a cluster of cells was localized within the distance of 10 μm (dashed line), which was set as the threshold to define 'basal' and 'inner' (luminal) cells. (**H**) Quantification of the proportion of epithelial cells in S/G2/M phase in basal and inner compartments in E13.5-E16.5 epithelial mammary rudiments. Sample sizes are as in (**E**). Data are presented as mean ± SD. The statistical significance was assessed using unpaired two-tailed Student's *t*-test with Bonferroni correction. ns, non-significant; *, p<0.05; ***, p<0.001; ****, p<0.0001.

The online version of this article includes the following source data for figure 2:

**Source data 1.** Source data of quantifications represented as graphs in *Figure 2B, C, E and H*.

## Basal-cell biased proliferation is not sufficient to drive initiation of branching

The observation that basal cell-biased proliferation occurred prior to onset of branching suggests that it might be a prerequisite for branching to occur. To further investigate the potential link between proliferation and initiation of branching, we took advantage of a mouse model that displays precocious branching, the Krt14-Eda mouse overexpressing Eda under the keratin 14 (Krt14) promoter (*Mustonen et al., 2003*). Eda and its epithelially-expressed receptor Edar regulate growth and branching of the embryonic and pubertal mammary gland (*Chang et al., 2009*; *Elo et al., 2017*; *Voutilainen et al., 2012*; *Voutilainen et al., 2015*; *Williams et al., 2022*). In Krt14-Eda embryos, mammary epithelial proliferation is increased, and branching is initiated already at E14.5 (*Voutilainen et al., 2012*).

To more closely examine the cellular alterations induced by Eda, we quantified the size, branch tip number, and proliferation status in Krt14-Eda embryos and their wild type littermates at E13.5 and E14.5. Mammary buds of Krt14-Eda embryos were significantly larger already at E13.5 (*Figure 3A and B*), and at E14.5, the volume was comparable to those of E16.5 wild type embryos (compare *Figure 3B* to *Figure 2B*, all mice in C57Bl/6 background). As reported (*Voutilainen et al., 2012*), branching was evident in Krt14-Eda embryos already at E14.5 (*Figure 3C*).

Further analysis of Fucci2a reporter expression in Krt14-Eda embryos at E13.5 and E14.5 revealed that the portion of S/G2/M cells was significantly higher in Krt14-Eda mice at both stages compared with wild type littermates (*Figure 3D* and *Figure 3—figure supplement 1A*). In addition, the basal cell-biased proliferation was evident already at E14.5 (but not yet at E13.5) in Krt14-Eda embryos (*Figure 3E*), similar to wild type mice at E15.5/E16.5 (*Figure 2H*). Since E14.5 Krt14-Eda mammary glands had similar characteristics to E16.5 wild type in terms of volume, elevated overall proliferation, and basal cell-biased proliferation, we next tested their ability to grow and branch in the mesenchyme-free 3D Matrigel culture. E14.5, but not E13.5, Krt14-Eda epithelia were able to branch, whereas epithelia isolated from wild type littermates expectedly failed to generate outgrowths (*Figure 3F and G*). We also analyzed Fucci2a reporter expression in *Eda*-/- mice (*Srivastava et al., 1997*) at E15.5 and E16.5. As we previously reported (*Voutilainen et al., 2012*), loss of *Eda* led to smaller glands and branching was delayed with most mammary glands being unbranched at E16.5 (*Figure 3H–J*), overall proliferation being also reduced, in particular at E16.5 (*Figure 3K* and *Figure 3—figure supplement 1B*), which together with the smaller anlage already at E13.5 and the slightly diminished cell size (*Figure 3—figure supplement 1C–E*) likely explains the smaller size of the E15.5-E16.5 *Eda*-/- mammary glands. However, the relative portion of S/G2/M cells in basal and inner cells (*Figure 3L* and *Figure 3—figure supplement 1B*) were similar between *Eda*-/- and wild type controls at both stages.

Next, we evaluated the branching ability by performing mesenchyme-free 3D culture. While nearly all E16.5 control epithelia gave rise to branched outgrowths, as expected, about half of *Eda*-/- epithelia failed to do so (*Figure 3M and N*). Collectively, these data indicate that initiation of the first branching events succeeds activation of proliferation, coordinated by the Eda signaling pathway, but is not its direct consequence.

## Salivary gland mesenchyme is rich in growth-promoting cues

Next, we shifted our focus to the regulation of the branching pattern, which is thought to be determined by mesenchymal cues (*Kratochwil, 1969*; *Sakakura et al., 1976*). To assess the influence of the mesenchyme, we performed heterotypic and heterochronic epithelial-mesenchymal recombination experiments between fluorescently labeled mammary and salivary gland tissues. Mammary epithelia and mesenchymes were isolated either at the quiescent bud stage (E13.5), or right after the bud had sprouted (E16.5); in addition to the primary mesenchyme, also mammary fat pad precursor tissue was micro-dissected from E16.5 embryos. Salivary gland tissues were isolated at E13.5, when the first branching events are evident and tissue separation is effortless. Homotypic recombinations were used as controls.

As previously reported (*Kratochwil, 1969*), E16.5 mammary ductal trees were far denser when cultured with salivary gland mesenchyme, and grew and branched at a faster rate than with any of the mammary mesenchymes tested (*Figure 4A*, top row). Of E13.5 mammary epithelia, majority (13 out of 18) did not survive in the salivary gland mesenchyme, and in the remaining ones, only traces of epithelial cells could be detected after 6 days of culture (*Figure 4A*, middle row). However, E13.5

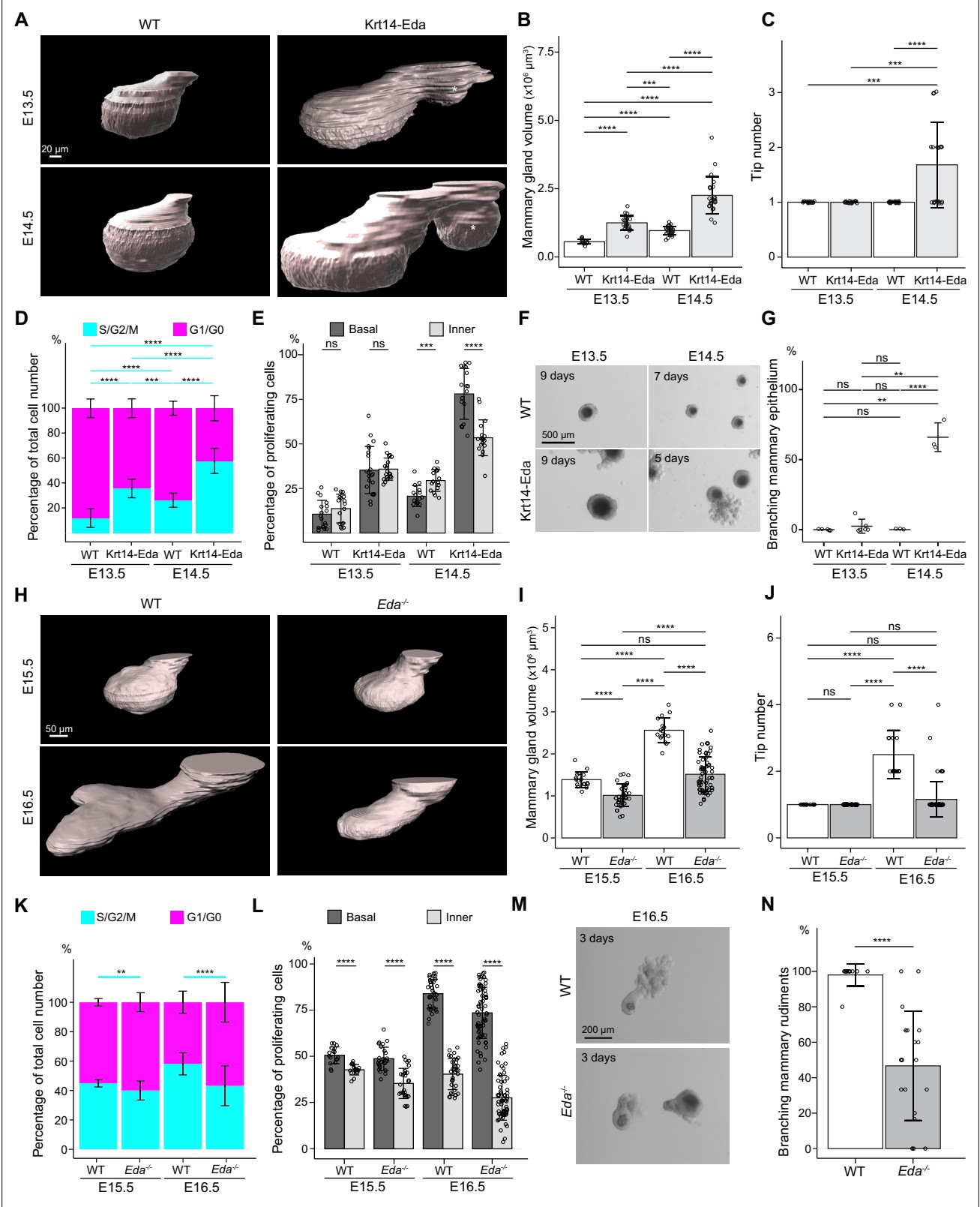

**Figure 3.** Basal-cell biased proliferation precedes, but is not sufficient to drive onset of branching. (**A**) Representative 3D surface rendering images of EpCAM-stained mammary glands of Krt14-Eda embryos and their wild type (WT) littermates at E13.5 and E14.5. Mammary gland 2 is shown. Ectopic mammary rudiments (asterisk) common in Krt14-Eda embryos were excluded from the analysis. Scale bar, 20 μm. (**B, C**) Quantification of mammary gland volume (**B**) and branching tip number (**C**) at E13.5 ($n_{WT}$ = 17, $n_{Krt14-Eda}$=21) and at E14.5 ($n_{WT}$ = 27 and $n_{Krt14-Eda}$=22). (**D, E**) Quantification of the

*Figure 3 continued on next page*

*Figure 3 continued*

proportions of mammary epithelial cells in S/G2/M and G1/G0 phases in the entire epithelium (**D**) and the proportions of mammary epithelial cells in S/G2/M phase in basal and inner compartments (**E**) in WT or Krt14-Eda embryos at E13.5 ($n_{WT}$ = 17 glands and in total 7714 cells from three embryos, $n_{Krt14-Eda}$=21 glands and in total 15,561 cells from 4 embryos) and E14.5 ($n_{WT}$ = 16 glands and in total 10,221 cells from 4 embryos, $n_{Krt14-Eda}$=18 glands and in total 10,520 cells from 5 embryos). (**F**) Representative images showing the growth of E13.5 and E14.5 Krt14-Eda and wild type littermate epithelial mammary rudiments in 3D Matrigel culture. Note branching in E14.5 Krt14-Eda mammary rudiments. Scale bar, 500 µm. (**G**) Quantification of branching mammary rudiments in 3D culture. Data are presented as percentage of branching mammary rudiments (mean ± SD) from a total of 5 (E13.5 WT), 6 (E13.5 Krt14-Eda), 3 (E14.5 WT) and 3 (E14.5 Krt14-Eda) independent experiments (each with minimum 5 rudiments in culture). (**H**) Representative 3D surface rendering images of EpCAM-stained E15.5 and E16.5 epithelial mammary rudiments of *Eda⁻ᐟ⁻* and wild type embryos. Mammary gland 2 is shown. Scale bar, 50 µm. (**I, J**) Quantification of epithelial mammary gland volume (**I**) and number of branching tips (**J**), at E15.5 ($n_{WT}$ = 17 and $n_{Eda-/-}$ = 33) and at E16.5 ($n_{WT}$ = 32 and $n_{Eda-/-}$ = 68). (**K, L**) Quantification of the proportions of mammary epithelial cells in S/G2/M or G1/G0 phases (**K**) and the proportions of mammary epithelial cells in S/G2/M phase in basal and inner compartments (**L**) in WT or *Eda⁻ᐟ⁻* embryos at E15.5 ($n_{WT}$ = 17 glands and in total 14,054 cells from 3 embryos, $n_{Eda-/-}$ = 27 glands and in total 21,986 cells from 5 embryos) and E16.5 ($n_{WT}$ = 34 glands and in total 72,279 cells from 3 embryos, $n_{Eda-/-}$ = 64 glands and in total 76,844 cells from 3 embryos). (**M**) Representative images showing E15.5 and E16.5 *Eda⁻ᐟ⁻* and wild type epithelial mammary rudiments in 3D culture after 3 days. Scale bar, 200 µm. (**N**) Quantification of branching mammary rudiments in 3D culture. Data are presented as percentage of branching mammary rudiments from a total of 10 WT and 19 *Eda⁻ᐟ⁻* E16.5 embryos (each with 3–6 rudiments in culture). Data are presented as mean ± SD. The statistical significance was assessed using unpaired two-tailed Student's *t*-test with Bonferroni correction, except Wilcoxon test with Bonferroni correction for (**C, G** and **J**). ns, non-significant; *, p<0.05; **, p<0.01; ***, p<0.001; ****, p<0.0001.

The online version of this article includes the following source data and figure supplement(s) for figure 3:

**Source data 1.** Source data of quantifications represented as graphs in *Figure 3B–E, G1–L and N*.

**Figure supplement 1.** The cellular dynamics of mammary epithelium in *Eda* gain-of-function and loss-of-function mouse models.

**Figure supplement 1—source data 1.** Source data of quantifications represented as graphs in *Figure 3—figure supplement 1C–E*.

---

mammary epithelia branched readily in combination with all mammary mesenchymes (*Figure 4A*, middle row), although their success rate was generally lower than that of E16.5 epithelia, as also previously reported (*Kratochwil, 1969*). In addition, we assessed the impact of mammary mesenchyme on salivary gland epithelium. Although the salivary gland epithelium usually survived, further growth and branching were minimal when cultured with any of the mammary mesenchymes, in stark contrast with homotypic control recombinants (*Figure 4A*, bottom row).

## Salivary gland mesenchyme does not alter the mode of branch point formation of the mammary epithelium

In principle, new branches can be generated by two different mechanisms: tip clefting/bifurcation or lateral (side) branching (*Lang et al., 2021*; *Myllymäki and Mikkola, 2019*). In the embryonic mammary gland, both events are common (*Lindström et al., 2022*) while the salivary gland branches by tip clefting only (*Wang et al., 2017*). Recent advances in imaging technologies have enabled time-lapse analysis of branching events in detail prompting us to perform live imaging of salivary and mammary epithelia recombined ex vivo with salivary gland mesenchyme (*Figure 4B*, *Figure 4—video 1* and *Figure 4—figure supplement 1*). Images were captured at 2 hr intervals, and branching events were traced and quantified from the time-lapse videos. Nearly all salivary gland branching events occurred by tip clefting (*Figure 4C*), as expected. Surprisingly, over 60% of mammary branching events were generated by lateral branching in either salivary mesenchyme or mammary mesenchyme with similar incidence, the latter finding being consistent with our previous report of ex vivo cultured mammary glands that did not undergo tissue separation prior to culture (*Lindström et al., 2022*; *Myllymäki et al., 2023*). We conclude that although salivary gland mesenchyme boosts growth of the mammary epithelium, the mode of branching is an intrinsic property of the mammary epithelium that is not altered by the growth-promoting salivary gland mesenchyme environment.

## Transcriptomic profiling of mammary and salivary gland mesenchymes identifies potential growth regulators

To identify the mesenchymal cues governing the differential growth characteristics of mammary and salivary gland epithelia, we performed transcriptomic profiling of five distinct tissues: E13.5 mammary mesenchyme surrounding the quiescent bud (E13.5 MM), E16.5 mammary mesenchyme surrounding the mammary sprout (E16.5 MM), E16.5 Fat pad precursor tissue (E16.5 FP), and E13.5 salivary gland mesenchyme (E13.5 SM) (*Figure 5A*). E13.5 non-mammary ventral skin mesenchyme (E13.5 VM) was

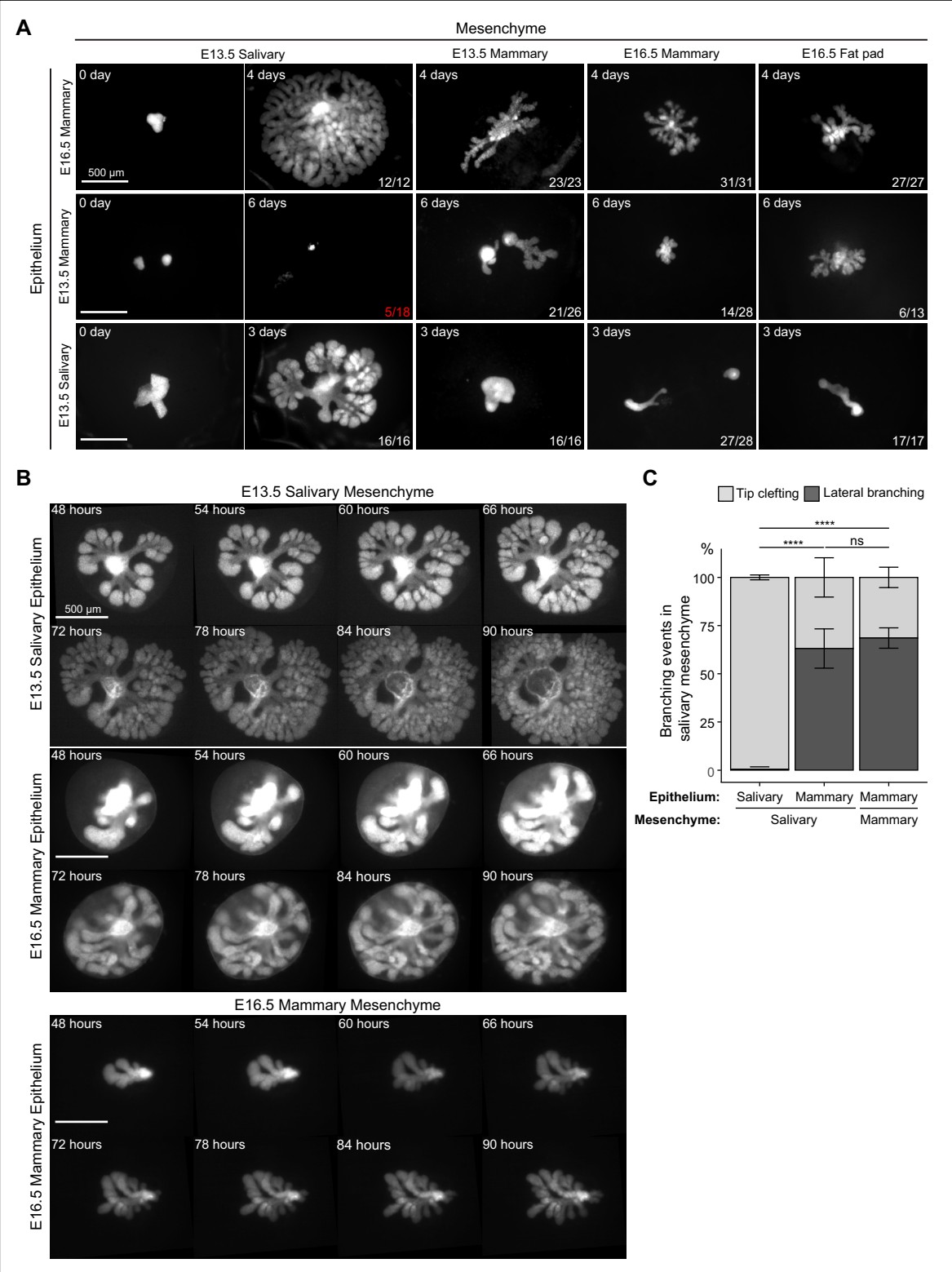

**Figure 4.** Mammary mesenchyme is indispensable for the branching ability of the mammary gland. Recombination experiments between micro-dissected mammary and salivary gland tissues using fluorescently labeled epithelia (see also *Figure 1*). (**A**) Representative images showing growth of the indicated epithelia recombined with distinct mesenchymes. Images were taken 0–6 days after culture as indicated in each figure. n in the lower right corner indicates growing recombinants out of those that survived, except for E13.5 mammary epithelium recombined with E13.5 salivary gland mesenchyme where it shows the number of survived recombinants/total recombinants (in red). In these recombinants, the epithelia never branched.

*Figure 4 continued on next page*

*Figure 4 continued*

Data were pooled from three to four independent experiments. Scale bars, 500 µm. (**B**) Captions of time-lapse live imaging series of explants consisting of E13.5 salivary epithelium or E16.5 mammary epithelium recombined with E13.5 salivary mesenchyme or E16.5 mammary mesenchyme. Images were captured every 2 hr starting 48 hr after recombination. The full video is provided as *Figure 4—video 1*. Scale bar, 500 µm. (**C**) Quantification of the branching events (lateral branching and tip clefting) from time-lapse videos. A pooled data from three independent experiments: in total of 239 branching events from 9 explants consisting of salivary epithelium and salivary mesenchyme, 159 branching events from 8 explants consisting of mammary epithelium and salivary gland mesenchyme and 40 branching events from 4 explants consisting of mammary epithelium and mammary gland mesenchyme were analyzed. Data are represented as mean ± SD and the statistical significance was assessed with unpaired two-tailed Student's *t*-test with Bonferroni correction. p values: ns, non-significant; ****, p<0.0001.

The online version of this article includes the following video, source data, and figure supplement(s) for figure 4:

**Source data 1.** Source data of quantifications represented as graphs in *Figure 4C*.

**Figure supplement 1.** Quality control of tissue separation and recombination.

**Figure 4—video 1.** Time-lapse live imaging showing the growth of E13.5 salivary epithelium (left) and E16.5 mammary epithelium (middle) in E13.5 salivary mesenchyme and E16.5 mammary epithelium in E16.5 mammary mesenchyme (right).

https://elifesciences.org/articles/93326/figures#fig4video1

also included to allow identification of mammary-specific transcriptomes. Five biological replicates for each tissue were sequenced.

The principal component analysis revealed that each group of samples were distinct from each other, although the E13.5 MM and E13.5 VM group quite close together (*Figure 5—figure supplement 1A*). To investigate the differences between the samples and assess the quality of the data, we performed pairwise comparisons and identified 51, 10, 54, 195, and 393 signature genes preferentially expressed in only one of the five sample sets (*Figure 5B* and *Supplementary file 1*). Among them, *Esr1* and *Ar* encoding estrogen and androgen receptors, respectively, were markers of E16.5 MM, while E16.5 FP was rich with adipogenesis markers such as *Aoc3, Adipoq, Cebpa, Fabp4, Lpl, Plin1, and Pparg* (*Menssen et al., 2011*). E13.5 SM-enriched genes *Nr5a2, Negr1, Klf14,* and *Satb2* have been identified as salivary mesenchyme markers by *Sekiguchi et al., 2020* using single-cell RNA sequencing. These data indicate that our RNA-Seq data represent well the transcriptomes of the designated tissues.

To understand the functional disparity between salivary and mammary mesenchymes in promoting epithelial growth and branching, we performed a Gene Ontology (GO) enrichment analysis for differentially expressed genes (DEGs) in Biological Processes (BP; *Figure 5C and D*). In total, 461 GOBP terms were shared among E13.5 MM, E16.5 MM and E16.5 FP when compared to E13.5 SM. Among the 461 shared GOBP terms, the top 10 most significantly enriched terms in each pairwise comparison resulted into 16 unique GOBP terms. Strikingly, of these, four were Wnt pathway related terms: canonical Wnt signaling pathway, regulation of canonical Wnt signaling pathway, negative regulation of Wnt signaling pathway, and negative regulation of canonical Wnt signaling pathway (*Figure 5D*).

To identify genes with the potential to regulate epithelial cell behaviors, we focused on DEGs encoding extracellular (secreted or membrane-bound) molecules (signaling molecules, signaling pathway inhibitors, extracellular matrix components) in biologically relevant pairwise comparisons (*Figure 5E*). Exclusion of lowly expressed genes led to the identification of 644 candidate genes (*Figure 5—figure supplement 1*). mFuzz cluster analysis (*Krull et al., 2019*) suggested that those genes could be further classified into 9 clusters based on their expression pattern across all the samples (*Figure 5F* and *Supplementary file 2*). Examination of the Wnt pathway related genes (as identified by GOBP enrichment analysis shown in *Figure 5D*) in these clusters revealed that altogether 12 out of 19 negative regulators of Wnt pathway were markers of clusters 1 and 3, including *Dkk2, Bmp2, Wnt11, Slc9a3r1, Grem1, Wif1, Tsku, Wnt5a, Dkk1, Notum, Sostdc1,* and *Cthrc1* (*Figure 5G*). Clusters 1 and 3 were characterized by genes displaying lower expression in E16.5 MM than E13.5 MM, and the lowest level in E13.5 SM (*Figure 5F*). Our tissue recombination experiments (*Figure 1B*) suggest that such expression pattern might represent potential growth suppressors. In other words, low expression of these negative regulators in salivary gland mesenchyme might enhance epithelial growth and branching, and in turn their higher expression in mammary mesenchyme might inhibit growth.

Clusters 2, 7, 8, and 9 were defined by genes such as *Hgf, Ltbp1, Tnc, and Postn,* with highest expression levels in one or more mammary-derived mesenchymes, highlighting them as best

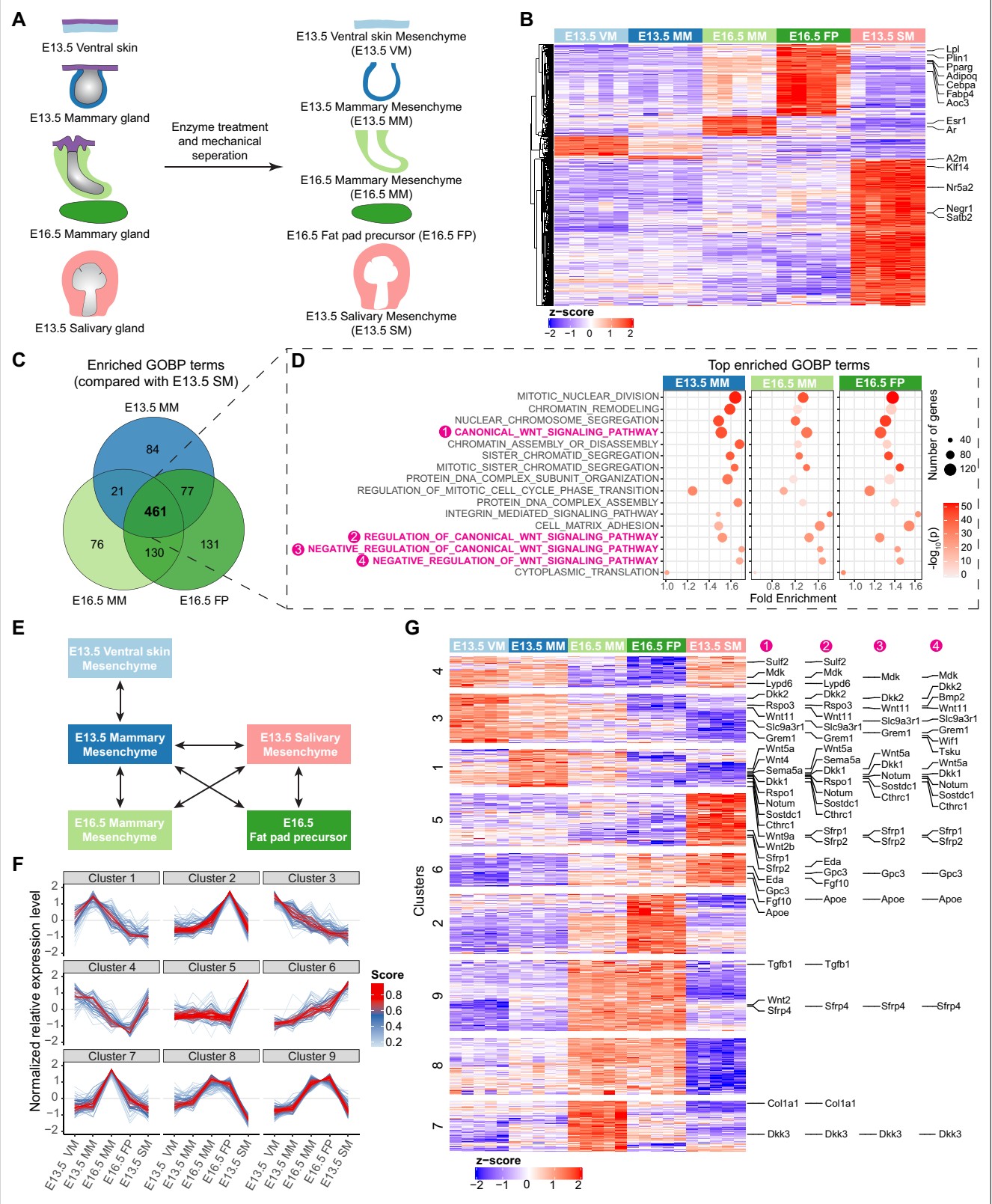

**Figure 5.** Transcriptomic analysis identifying mesenchymal signals potentially regulating epithelial growth. (**A**) A scheme illustrating the tissues isolated for RNA-Seq analysis. (**B**) Heatmap showing the expression of the identified marker genes (with a threshold of average of normalized expression value in each group ≥100, fold change ≥2 and adjusted p-value <0.05) in different mesenchymes using the z-score of log2-transformed normalized expression value (also see **Supplementary file 1**). (**C**) Venn diagram showing 461 enriched Gene Ontology Biological Process (GOBP) terms shared among E13.5

*Figure 5 continued on next page*

*Figure 5 continued*

mammary mesenchyme (MM), E16.5 MM and E16.5 fat pad (FP) when compared to E13.5 salivary gland mesenchymes (SM) separately. (**D**) Top 10 (among the 461 shared terms) of the most significantly enriched GOBP terms in each comparison resulted in 16 distinct terms in total. Four out of 16 terms were related to Wnt signaling pathway (in magenta). (**E**) A scheme illustrating the pair-wise comparisons used to identify the genes with the potential to regulate epithelial growth. Altogether 644 genes encoding extracellular matrix proteins and ligands with average of normalized expression value in each group ≥200, fold change ≥1.5 and adjusted p-value <0.05 were identified. (**F**) mFuzz cluster analysis of the genes identified in (**E**) (also see *Supplementary file 2*). (**G**) Heatmap showing the expression of genes identified in (**E**) using the z-score of log2-transformed normalized expression value. The clusters were defined by mFuzz shown in (**F**). The genes within the Wnt related GOBP terms identified in (**D**) are indicated accordingly in the right.

The online version of this article includes the following figure supplement(s) for figure 5:

**Figure supplement 1.** Transcriptomic profiling of different mesenchymes.

candidates to possess mammary-specific functions, for example in regulation of sprouting or epithelial cell differentiation. On the other hand, the clusters 5 (e.g. *Adam10*, *Adamts1*, *Bmp1*, *Bmp7*) and 6 (e.g. *Fgf10*, *Igf1*, *Igf2*, and *Eda*) genes have highest expression levels in E13.5 SM, indicating a potential role as drivers of epithelial growth. This fits well with the known roles of Eda and Fgf10 in salivary and mammary gland development (*Häärä et al., 2011*; *Lindström et al., 2022*; *Prochazkova et al., 2018*; *Rivetti et al., 2020*; *Voutilainen et al., 2012*). One distinction between cluster 5 and 6 genes is that in the mammary gland, cluster 5 genes show invariable expression levels across all mammary mesenchymes, whereas cluster 6 genes show highest expression level in the fat pad where branching occurs. This increases the likelihood that cluster 6, rather than cluster 5, genes might be physiologically important, paracrine growth regulators of the mammary epithelium.

## Wnt-activated mesenchyme promotes growth of the mammary epithelium

The transcriptomic analysis suggests that one significant difference between salivary and mammary mesenchymes is the Wnt pathway. Gene set variation analysis (GSVA) suggested that the Wnt signaling signature was higher in E13.5 SM compared to all mammary mesenchymes (*Figure 5—figure supplement 1B*), which is consistent with the high expression of Wnt inhibitors in the mammary mesenchyme. In the RNA-Seq dataset, *Axin2* mRNA level, often used as a readout of canonical Wnt activity, were significantly higher in the salivary gland mesenchyme compared to the E16.5 fat pad where mammary branching takes place (*Figure 5—figure supplement 1C*). Expression of the TCF/LEF:H2B-GFP Wnt reporter (*Ferrer-Vaquer et al., 2010*) was also higher in E13.5 salivary gland mesenchyme compared to the E16.5 mammary mesenchyme (*Figure 6A and B*). Moreover, we have previously shown that suppression of mesenchymal Wnt activity in developing salivary glands compromises growth of the salivary gland (*Häärä et al., 2011*). Together, these findings prompted us to ask whether low levels of mesenchymal Wnt activity could limit the growth of the mammary epithelium. To answer this question experimentally, we aimed to activate Wnt signaling by stabilizing β-catenin (encoded by *Ctnnb1*) in the mesenchyme by crossing *Twist2*^Cre+/- mice with those harboring exon3 –floxed *Ctnnb1* (*Ctnnb1*^lox(ex3)/lox(ex3) mouse) (*Harada et al., 1999*). However, this led to embryonic lethality already at E12.5, in line with previous reports (*Tran et al., 2010*). Therefore, we chose the tissue recombination approach where E13.5 wild type mammary buds were recombined with E13.5 mammary mesenchyme dissected either from control (*Ctnnb1*^+/+) or *Ctnnb1*^lox(ex3)/+ embryos, followed by adeno-associated virus (AAV8) –mediated gene transduction as a means to deliver Cre recombinase (*Lan and Mikkola, 2020*; *Figure 6C*). As a result, Wnt signaling was activated in the mesenchymal cells only. Quantification of tissue recombinants transduced with AAV8-Cre revealed that wild type mammary epithelia cultured on mammary mesenchyme from *Ctnnb1*^lox(ex3)/+ embryos had significantly more ductal tips than those cultured on control mammary mesenchyme (*Figure 6D and E*). These data indicate that low level of mesenchymal Wnt signaling activity limits expansion and branching of the mammary epithelium.

Next, we asked which paracrine factors could regulate epithelial growth downstream of mesenchymal Wnt signaling. First, we explored a publicly available RNA-Seq dataset (*Wang et al., 2021*; *Figure 6F*) which compared gene expression levels in wild type and β-catenin deficient mammary fibroblasts cultured with or without Wnt3a protein, and narrowed our analysis on cluster 5 and 6 genes identified in the mFuzz analysis (*Figure 5F* and *Supplementary file 2*). These genes displayed

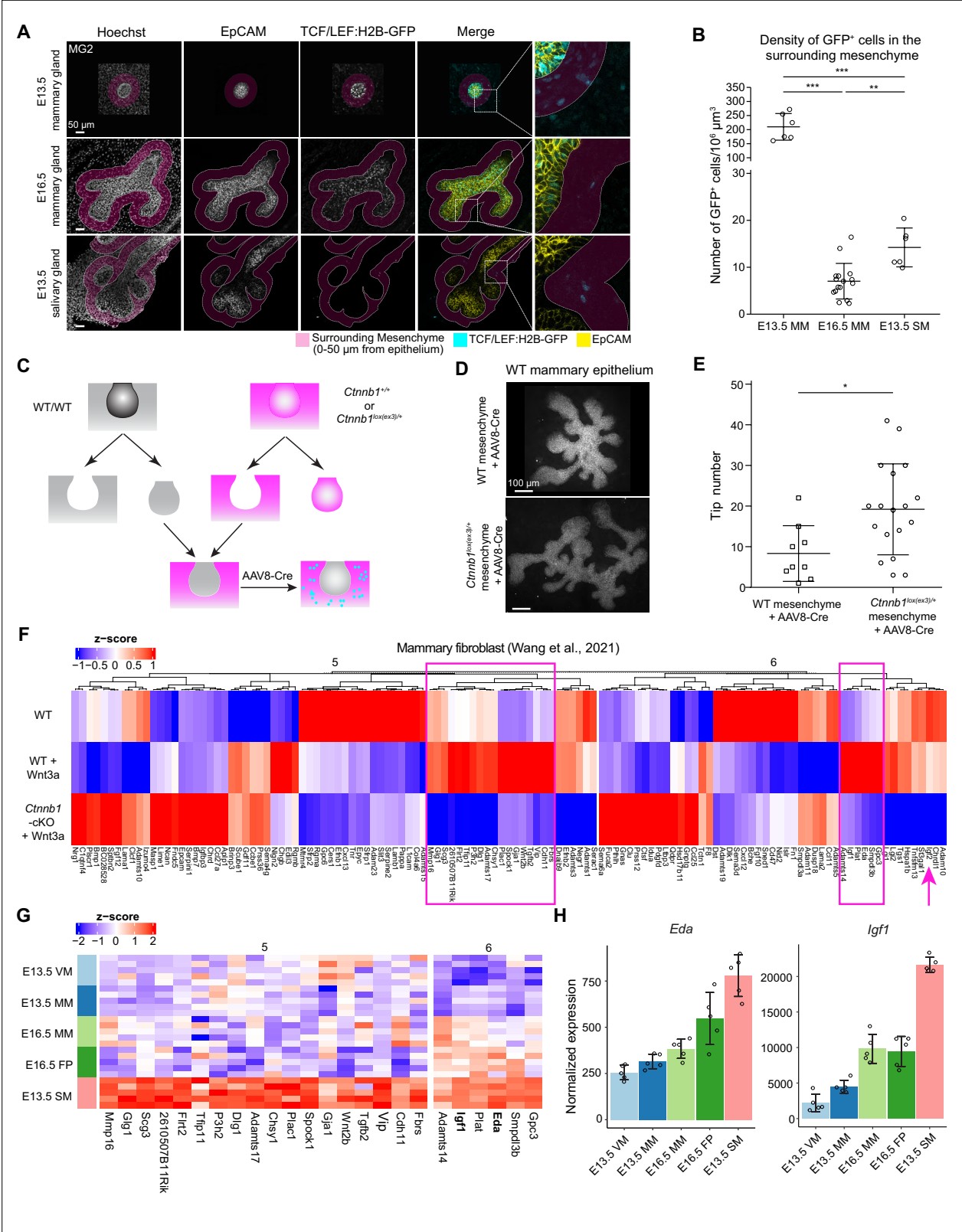

**Figure 6.** Wnt-activated mesenchyme promotes growth of the mammary epithelium. (**A**) Confocal optical sections of whole mount EpCAM-stained tissues expressing TCF/LEF:H2B-GFP Wnt reporter from E13.5 and E16.5 mammary glands and E13.5 salivary glands. The mesenchyme within 0–50 μm distance from epithelia in 3D was labeled as magenta. Scale bars, 50 μm. (**B**) Quantification of the density of mesenchymal Wnt reporter-expressing (GFP+) cells within 0–50 μm distance from the epithelium. Altogether, 6 and 16 mammary gland 2 from three E13.5 embryos and eight E16.5 embryos,

*Figure 6 continued on next page*

*Figure 6 continued*

respectively, and 6 salivary glands from three E13.5 embryos were analyzed. Data are presented as mean ± SD. Statistical significance was assessed using unpaired two-tailed Student's *t*-test with Bonferroni correction. **, p<0.01; ***, p<0.001. (**C**) A scheme illustrating the experimental design for mesenchymal activation of Wnt/ß-catenin signaling activity. (**D**) Representative images showing EpCAM stained wild type mammary epithelia after 6 days culture in wild type or *Ctnnb1*$^{lox(ex3)/+}$ mesenchyme infected with AAV8-Cre virus during the first 48 hr. (**E**) Quantification of the number of branching tips of wild type mammary epithelia recombined with wild type or *Ctnnb1*$^{lox(ex3)/+}$ mesenchyme after 6 days of culture. Data are presented as mean ± SD (n=9 and 18 for WT and *Ctnnb1*$^{lox(ex3)/+}$ mesenchyme, respectively) and represented from three independent experiments. Statistical significance was assessed using unpaired two-tailed Student's *t*-test. *, p<0.05. (**F**) Unsupervised cluster of heatmap showing the expression of cluster 5 and 6 genes identified by mFuzz analysis (see *Figure 5F*) in a published dataset (*Wang et al., 2021*) that compared gene expression levels in wild type and *β-catenin* deficient mammary fibroblasts cultured with or without Wnt3a protein. Data are shown as z-score of log2-transformed normalized expression values. Two subsets of potential mesenchymal Wnt target genes identified are marked (box in magenta). In addition, *Igf2* is highlighted with an arrow. (**G**) Heatmap showing the expression of the candidate genes from (**F**) in different mesenchymes of the RNA-Seq data. Data are shown as z-score of log2-transformed normalized expression values. (**H**) Graphs representing mRNA expression of *Eda* and *Igf1* as measured by RNA-Seq. Data are presented as normalized expression values (mean ± SD). Each dot represents one biological replicate.

The online version of this article includes the following source data and figure supplement(s) for figure 6:

**Source data 1.** Source data of quantifications represented as graphs in *Figure 6B, E and H*.

**Figure supplement 1.** Expression of IGF pathway genes in the mesenchymal tissues.

opposite expression patterns to genes in clusters 1 and 3, and hence were expected to positively regulate epithelial growth (*Figure 5F and G*). The analysis revealed that the expression of most of the cluster 5 and 6 genes was altered in mammary fibroblasts upon manipulation of Wnt signaling activity (*Figure 6F*). Focusing on genes upregulated by Wnt3a in wild type, but not in β-catenin deficient fibroblast led to the identification of 18 and 5 candidate genes in clusters 5 and 6, respectively, *Eda* and *Igf1* being amongst them, while *Igf2* was somewhat decreased by the Wnt treatment (*Figure 6F–H*). We have previously identified *Eda* as a gene downstream of Wnt pathway in the salivary gland mesenchyme (*Häärä et al., 2011*), validating our analysis pipeline.

## IGF-1R is required for embryonic mammary gland development and branching morphogenesis

IGF-1 is well known for its role in growth control and, similar to other tissues, it functions as an important local mediator of the growth hormone in pubertal mammary glands (*Kleinberg and Ruan, 2008*; *Richards et al., 2004*; *Wood et al., 2000*). However, the role of the IGF-1 pathway in embryonic mammary gland development has not been explored, apart from one study reporting the smaller size of the E14 mammary bud in IGF-1R-deficient embryos (*Heckman et al., 2007*). Analyses of the known secreted components of the IGF pathway revealed that many of them were differentially expressed between salivary and mammary gland mesenchymes (*Figure 6—figure supplement 1*), the most striking being *Igf1* and pregnancy-associated plasma protein-A (*Pappa*), a zinc metalloproteinase that promotes IGF-1 signaling through cleavage of the inhibitory Igf-binding proteins (IGFBPs) (*Conover and Oxvig, 2018*). *Pappa* was also identified as a cluster 5 gene in the mFuzz analysis (*Figure 5—figure supplement 1*). To functionally test the effect of IGF-1 on embryonic mammary gland growth, we performed ex vivo culture of E16.5 mammary glands and treated the explants for 3 days with moderate levels of recombinant IGF-1 or vehicle (*Figure 7A*). Quantification of branch tip number showed that IGF-1 significantly increased expansion of the mammary epithelium (*Figure 7B*).

To assess the function of IGF-1 in vivo, we examined mammary gland development in embryos deficient for *Igf1r*, the obligate cognate receptor of Igf1 (*Dupont and Holzenberger, 2003*; *LeRoith et al., 2021*). As previously reported (*Liu et al., 1993*), *Igf1r*$^{-/-}$ embryos were significantly smaller compared with wild type littermates (*Igf1r*$^{+/+}$) (*Figure 7C*). At E16.5, the anterior glands of littermate control embryos had sprouted. Small outgrowths were also observed in *Igf1r*$^{-/-}$ embryos, with the exception of mammary gland 3 that was consistently absent (*Figure 7D*). At E18.5, growth and branching was severely compromised in the *Igf1r*$^{-/-}$ embryos, verified by quantification of the epithelial area of the mammary gland and the ductal tip number of mammary glands 1–4 at E18.5 (*Figure 7—figure supplement 1A and B*). To avoid biases caused by the conspicuously smaller size of the *Igf1r*$^{-/-}$ embryos (*Holzenberger et al., 2003*; *Liu et al., 1993*), we normalized the data to the body weight (*Figure 7E and F*). The normalized values revealed that the mammary gland area and tip numbers were significantly reduced in *Igf1r*$^{-/-}$ embryos compared to controls. There was no significant

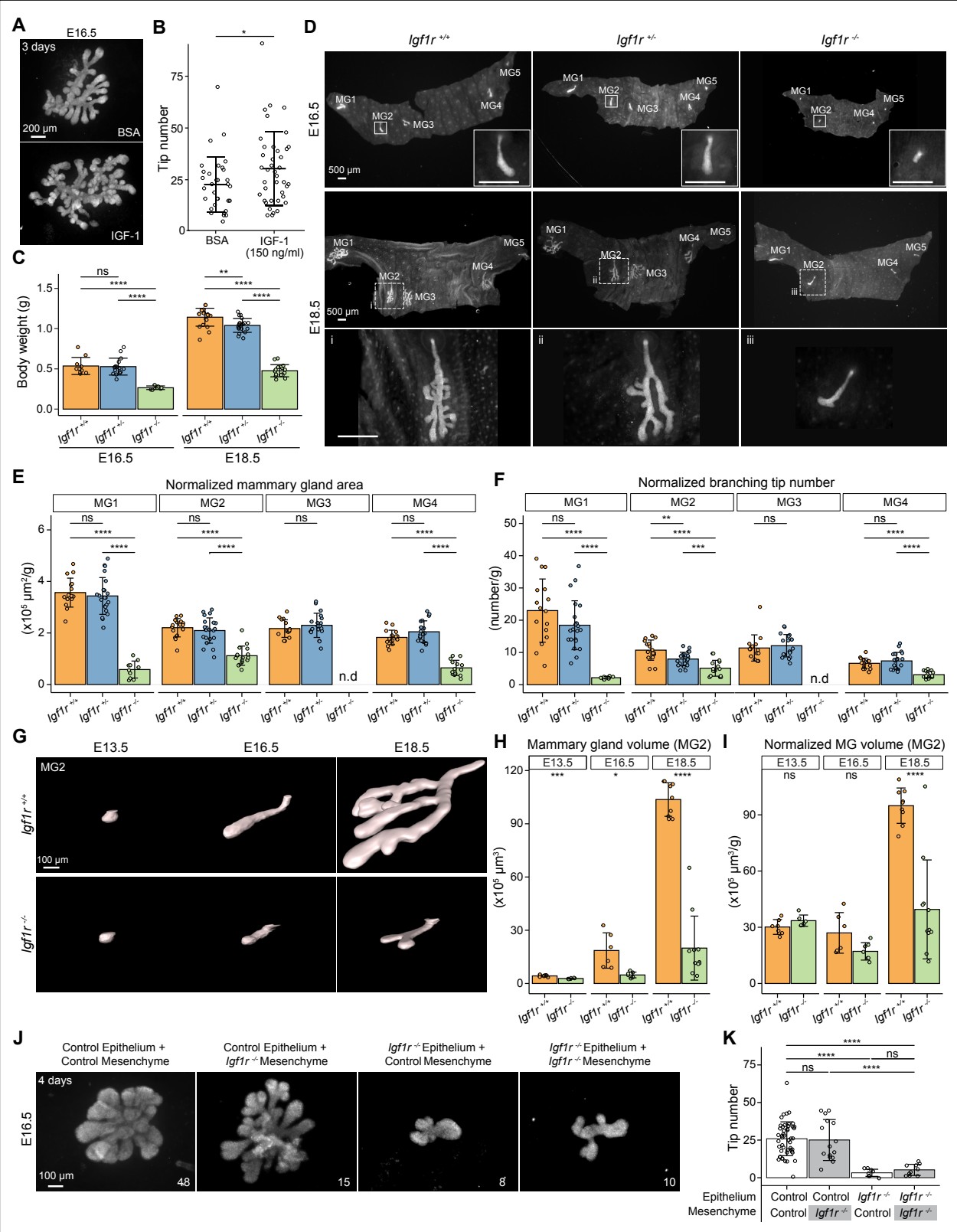

**Figure 7.** IGF-1R is required for embryonic mammary gland development and branching morphogenesis. (**A**) Representative images of E16.5 Krt14-Cre;Rosa26^mTmG/+ mammary glands cultured ex vivo for three days in the presence of 150 ng/ml recombinant IGF-1 or vehicle (BSA). Scale bar, 200 μm. (**B**) Quantification of the number of branching tips in vehicle (n=33) and IGF-1 treated (n=40) mammary gland explants. Data are pooled from five independent experiments and presented as mean ± SD. (**C**) Body weight of *Igf1r* ^+/+^, *Igf1r* ^+/-^and *Igf1r* ^-/-^ embryos at E16.5 (n_*Igf1r+/+*_=10, n_*Igf1r+/-*_=16,

*Figure 7 continued on next page*

*Figure 7 continued*

n$_{Igf1r-/-}$=7), and E18.5 (n$_{Igf1r+/+}$=20, n$_{Igf1r+/-}$=20, n$_{Igf1r-/-}$=17). (**D**) Representative images of EpCAM-stained ventral skin including mammary glands (MG) 1–5 from *Igf1r* $^{+/+}$, *Igf1r* $^{+/-}$ and *Igf1r* $^{-/-}$ female embryos at E16.5, and E18.5. Note absence of MG3 in *Igf1r* $^{-/-}$ embryos. Magnifications show mammary gland 2. Scale bars, 500 µm. (**E, F**) Quantification of mammary gland area (**E**) and number of branch tips (**F**) normalized to body weight in *Igf1r* $^{+/+}$, *Igf1r* $^{+/-}$ and *Igf1r* $^{-/-}$ embryos at E18.5. MG5 was often lost during dissection and therefore was not included in the analysis. n.d, not detected. (**G**) Representative 3D surface rendering images of EpCAM-stained mammary gland 2 from *Igf1r* $^{+/+}$ and *Igf1r* $^{-/-}$ embryos at E13.5 (n$_{Igf1r+/+}$=7, n$_{Igf1r-/-}$=6), E16.5 (n$_{Igf1r+/+}$=6, n$_{Igf1r-/-}$=7), and E18.5 (n$_{Igf1r+/+}$=9, n$_{Igf1r-/-}$=11), based on 3D confocal imaging. Scale bar, 100 µm. (**H–I**), Quantification of epithelial mammary gland volume (**H**) and volume normalized with body weight (**I**). Data are presented as mean ± SD. (**J, K**) Representative images (**J**) showing the growth of E16.5 mammary epithelia isolated from control (*Igf1r* $^{+/+}$ or *Igf1r* $^{+/-}$) or *Igf1r* $^{-/-}$ embryos recombined with E16.5 mammary mesenchyme from control or *Igf1r* $^{-/-}$ embryos, as indicated in each figure. Explants were cultured for 4 days and the epithelium visualized with EpCAM staining. Quantifications are shown in (**K**). Scale bar, 100 µm. Data are pooled from 6 independent experiments and presented as mean ± SD. n is indicated in the right corner of each image in (**J**). Statistical significances were assessed using unpaired two-tailed Student's *t*-test for (**A**) or unpaired two-tailed Student's *t*-test with Bonferroni correction for (**C, E, F, H, I** and **K**). ns, non-significant; *, p<0.05; **, p<0.01; ***, p<0.001; ****, p<0.0001.

The online version of this article includes the following source data and figure supplement(s) for figure 7:

**Source data 1.** Source data of quantifications represented as graphs in *Figure 7B, C, E, F, H, I and K*.

**Figure supplement 1.** Impact of *Igf1r* deficiency on mammary gland and salivary gland growth and branching.

**Figure supplement 1—source data 1.** Source data of quantifications represented as graphs in *Figure 7-figure supplement 1A-C*.

difference between *Igf1r* $^{+/-}$ and *Igfr1* $^{+/+}$ embryos, except that the number of tips in mammary gland 2 was reduced in *Igf1r* $^{+/-}$ embryos (*Figure 7F*). Analysis of E13.5 embryos revealed that mammary rudiment 3 was absent in *Igf1r* $^{-/-}$ embryos already early on (*Figure 7—figure supplement 1C and D*). Quantification of the epithelial volume of mammary gland 2 from *Igf1r* $^{-/-}$ and *Igfr1* $^{+/+}$ embryos at E13.5, E16.5, and E18.5 (*Figure 7G–I*) confirmed the significantly reduced size of the mammary anlage of *Igf1r* $^{-/-}$ embryos, the defect becoming notably pronounced at later developmental stages (*Figure 7H*). Importantly, normalization of the mammary gland volume to body weight revealed no difference between the genotypes at the bud stage, yet a progressive defect was evident from E16.5 onward, upon onset of branching morphogenesis (*Figure 7I*). In addition, we examined the developing salivary glands at E13.5, E16.5 and E18.5. In stark contrast to the mammary gland, the salivary glands of E16.5 and E18.5 *Igf1r* $^{-/-}$ embryos were highly branched although smaller (*Figure 7—figure supplement 1E*), paralleling the overall growth defect of the mutant embryos (*Figure 7C*).

The *Igf1r*-deficient mouse used in the current study is a constitutive gene deletion model, and hence the phenotype could result from lack of IGF-1R signaling in the epithelium, the mesenchyme, or both. To address this question, we conducted tissue recombination experiments involving mammary epithelial and mesenchymal tissues isolated from E16.5 control (*Igf1r* $^{+/+}$ or *Igf1r* $^{+/-}$) and *Igf1r* $^{-/-}$ embryos. Absence of *Igf1r* in the mesenchyme did not impair growth and branching of the control epithelium (*Figure 7J and K*), while *Igf1r* $^{-/-}$ epithelium failed to grow even if recombined with control mesenchyme, indicating that epithelial *Igf1r* deficiency is the primary cause of the branching defects observed in *Igf1r* $^{-/-}$ embryos.

Collectively, these data show that embryonic mammary gland development is exceptionally sensitive to loss of IGF-1/IGF-1R signaling, as shown by the complete absence of mammary bud 3 and the specific growth and branching impairment during late embryogenesis.

## Discussion

In this study, we explored the fundamental principles of epithelial-mesenchymal tissue interactions guiding embryonic mammary gland development. Our findings reveal that while both the timing and type of branching events are intrinsic properties of the mammary epithelium, mammary-specific mesenchymal signals are crucial for the acquisition of the branching capacity. Importantly, we demonstrate that salivary gland mesenchyme could only promote the growth of the later stage (E16.5) mammary epithelium without changing the branching regime. Transcriptomic profiling and experimental evidence indicate that mesenchymal Wnt signaling and Igf1 downstream of it are critical regulators promoting expansion of the mammary gland epithelium and contribute to the differences in growth-promoting capacity of the mammary and salivary mesenchymes. Other pathways are also involved, as several signaling molecules known to regulate growth, such as *Eda* and *Fgf10* (*Jaskoll*

*et al., 2005*; *Lindfors et al., 2013*; *Lindström et al., 2022*), were differentially expressed between salivary and mammary gland mesenchymes.

Two important events occur before initiation of mammary gland branching: exit from quiescence and obtaining outgrowth capacity. Krt14-Eda data suggest that these two phenomena are likely coordinated, in part through Eda signaling. Interestingly, cells in the basal layer are more proliferative initially, unlike during later embryogenesis when branching is ongoing (*Myllymäki et al., 2023*). Our observation that proliferation is specifically activated in the basal layer prior to branching seems to support the previous hypothesis that proliferation and lineage segregation may be linked to drive onset of branching (*Inman et al., 2015*; *Lilja et al., 2018*), but further studies will be needed to address this question. The fact that E13.5 Krt14-Eda and E15.5 wild type mammary epithelia fail to grow and branch in 3D culture despite the high proliferation rate, implies that additional factors are required to acquire branching capacity. This is in line with our recent study showing that inhibition of cell proliferation does not prevent branch point generation per se, though new cells are evidently needed as building blocks for further ductal growth (*Myllymäki et al., 2023*). Instead, cell motility is critical for branch point formation in the mammary gland (*Myllymäki et al., 2023*), as well as in other branching organs (*Chi et al., 2009*; *Kim et al., 2013*; *Nakanishi et al., 1987*). Accordingly, we observed significantly increased expression of cell migration promoting genes such as *Cdh11* (encoding Cadherin 11) and *Tnc* (encoding Tenascin C) (*Andrews et al., 2012*; *Midwood et al., 2016*) in E16.5 mesenchyme compared to E13.5 (*Supplementary file 2*).

Epithelial-mesenchymal tissue recombination experiments performed mainly in the 50s to 70s using different branched organs, including the lung, kidney, and salivary gland, have disclosed the dominant role of the mesenchyme in branch patterning (*Alescio and Cassini, 1962*; *Alescio and Di Michele, 1968*; *Alescio and Piperno, 1967*; *Grostein, 1953*; *Iwai et al., 1998*; *Kispert et al., 1996*; *Lawson, 1974*; *Lawson, 1983*), a conclusion confirmed also by detailed branch pattern analyses of heterotypic kidney and lung tissue (*Lin et al., 2003*). Similarly, recombination experiments between mammary epithelium and salivary gland mesenchyme (*Kratochwil, 1969*; *Sakakura et al., 1976*) laid the foundation for our current understanding on the instructive role of the mesenchyme in mammary gland branching morphogenesis. However, at the time, time-lapse imaging was not feasible precluding a comprehensive investigation of the dynamic branching process. Advances in imaging may explain our contrasting result. That is, our data clearly demonstrate that although the density and growth rate of the mammary ductal tree were greatly enhanced by the salivary gland mesenchyme, the type of branch point formation was not. This observation suggests that mammary epithelium itself carries the instructions dictating the mode of branching involving both lateral branching and tip bifurcations. This conclusion is further supported by our recent study showing that isolated E16.5 mammary epithelia retain bimodal branching also in the mesenchyme-free 3D organoid culture (*Myllymäki et al., 2023*). Evidently, further studies are required to elucidate which properties of the mammary epithelium enable its bimodal branching behavior. It is worth noting that certain mesenchymal factors, such as Ltbp1, began transitioning towards epithelium-specific expression around E16.5 (*Chandramouli et al., 2013*). Exploring the potential impact of these factors on the self-instructed branching capacity of the mammary epithelium could yield valuable insights.

In contrast to the mode of branching, growth rate and density of the mammary ductal tree was grossly altered by the salivary gland mesenchyme implying an important role for paracrine factors in these processes. This, together with the failure of the salivary epithelium to grow in mammary gland mesenchyme indicate that the mammary gland mesenchyme is either poor in growth-promoting cues and/or rich in growth-inhibitory cues. Our transcriptomic profiling suggest that it may be both. Growth factors like *Fgf10*, *Eda*, and *Igf1* were expressed at higher levels in the salivary gland mesenchyme, while the RNA-Seq data indicated that low level of mesenchymal Wnt activity, likely in part due to high levels of Wnt inhibitors, may restrict mammary gland expansion. Mesenchymal Wnt activity is critical for the early specification of the mammary mesenchyme (*Hiremath et al., 2012*), but its function beyond the bud stage is largely unknown. The *Axin2* and Wnt reporter expression analyses indicate that mesenchymal Wnt activity is reduced by the time branching begins. In addition, our experimental data revealed that growth and branching of the mammary gland was enhanced by mesenchymal activation of Wnt/β-catenin signaling activity. Previous studies have shown that an excess of Wnt ligands promotes growth of the embryonic mammary epithelium but the primary target tissue was unknown (*Cunha and Hom, 1996*; *Voutilainen et al., 2012*). Our results suggest that this could be (in part) an

indirect effect, due to augmented mesenchymal Wnt signaling activity. This hypothesis is consistent with our recent study demonstrating that forced stabilization of epithelial β-catenin compromises branching of the embryonic mammary gland (*Satta et al., 2023*).

The IGF-1/IGF-1R signaling pathway has a critical role in the coordinated regulation of body growth downstream of the pituitary growth hormone (*LeRoith et al., 2021*; *Streck et al., 1992*). In its absence, the size of the organs is also proportionally reduced (*LeRoith et al., 2021*; *Powell-Braxton et al., 1993*). Here we show that the embryonic mammary gland is particularly sensitive to *Igf1r* deficiency, mammary gland 3 failing to develop at all. These data suggest that the role of IGF-1R during mammary gland development, particularly in the branching morphogenesis, extends beyond its general growth promoting function during embryonic development. The reason for this is currently unknown but one possibility is that the availability of active IGFs in mammary gland mesenchyme is limited to begin with, due to low expression of *Pappa*. Normally, the IGFs exist in the form of binary complexes with IGFBPs, and PAPPA degrades IGFBPs, increasing the bioavailable fraction of IGFs thereby promoting activation of IGF-1R (*LeRoith et al., 2021*). Due to the functional redundancy between IGF-1 and IGF-2 in IGF-1R activation, we cannot exclude the potential role of IGF-2 in promoting mammary gland branching via IGF-1R. However, as *Igf2* expression was suppressed by Wnt3a in mammary fibroblast, we find it unlikely that IGF-2 mediates the Wnt-IGF-1R crosstalk.

In conclusion, our findings provide valuable insights into the growth control of the mammary gland and the transcriptomic profiling of different mesenchymes as a novel resource for investigating the mesenchymal contribution in organ development. Intriguingly, we found that heterochronic mammary mesenchyme did not advance/delay the timing of epithelial outgrowth and branching, indicating that mechanisms intrinsic to the mammary epithelium govern these processes. Yet, mammary-specific mesenchyme was indispensable for branching to occur, suggesting that mammary mesenchyme may provide permissive cues that allow the mammary bud to exit quiescence and become competent to respond to mitogenic cues. Parathyroid hormone like hormone (Pthlh, also known as Pthrp) signaling may play a critical role here: deletion of the mesenchymally expressed receptor *Pthr1* or the epithelially expressed ligand halts mammary gland development at E15.5-E16.5, prior to onset of branching (*Wysolmerski et al., 1998*). However, the downstream targets of Pthr1 are incompletely understood, but both Wnt and bone morphogenetic protein (Bmp) pathways are involved (*Hens et al., 2007*; *Hiremath et al., 2012*). In addition, the transcriptomic and epigenetic changes taking place in the mammary epithelium between the quiescent bud stage and growth competent sprout are currently unknown. Uncovering how mammary epithelial cells acquire their remarkable growth potential and identification of the underlying mesenchymal cues are fascinating avenues for future research with implications to our understanding of basic mammary gland biology, as well as breast cancer.

# Materials and methods

## Key resources table

| Reagent type (species) or resource | Designation | Source or reference | Identifiers | Additional information |
|---|---|---|---|---|
| Antibody | rat anti-mouse CD326 (EpCAM), monoclonal | BD Pharmingen | Cat# 552370; RRID:AB_394370 | 1:500 |
| Antibody | rabbit anti-mouse Krt14, polyclonal | Thermo Fisher Scientific (Lab Vision) | Cat# RB-9020-P; RRID:AB_149790 | 1:500 |
| Antibody | rabbit anti-cleaved Caspase-3, polyclonal | Cell Signaling Technology | Cat# 9661; RRID:AB_2341188 | 1:500 |
| Antibody | Alexa Fluor 488-conjugated Donkey anti-Rat secondary antibody, polyclonal | Thermo Fisher Scientific | Cat# A-21208; RRID:AB_2535794 | 1:500 |
| Antibody | Alexa Fluor 647-conjugated Donkey anti-Rat secondary antibody, polyclonal | Thermo Fisher Scientific | Cat# A48272; RRID:AB_2893138 | 1:500 |
| Peptide, recombinant protein | Mouse IGF-1 | R&D systems | 791 MG | 150 ng/ml |
| Strain, strain background (*Mus musculus*, C57/Bl6) | Krt14-Eda | PMID:12812793 | | |
| Strain, strain background (*Mus musculus*, C57/Bl6) | Krt14-Cre | PMID:1508351815083518 | | |

*Continued on next page*

*Continued*

| Reagent type (species) or resource | Designation | Source or reference | Identifiers | Additional information |
|---|---|---|---|---|
| Strain, strain background (*Mus musculus,*) | *Eda⁻/⁻* | The Jackson Laboratory | Strain #:000314; RRID:IMSR_JAX:000314 | |
| Strain, strain background (*Mus musculus, C57/Bl6*) | *Rosa26^Fucci2a flox/Fucci2a flox* | EMMA | EMMA:08395; RRID:IMSR_EM:08395 | The original strain was bred with Pgk1-cre before using in this study. |
| Strain, strain background (*Mus musculus, C57/Bl6*) | Pgk1-cre | The Jackson Laboratory | Strain #:020811; RRID:IMSR_JAX:020811 | |
| Strain, strain background (*Mus musculus, ICR*) | *Rosa26^mTmG* | The Jackson Laboratory | Strain #:007576; RRID:IMSR_JAX:007576 | |
| Strain, strain background (*Mus musculus, mix*) | *Rosa26^mGFP/mTmG* | This paper | | Obtained by breeding mTmG mouse with Pgk1-cre |
| Strain, strain background (*Mus musculus, C57/Bl6*) | *Ctnnb1^lox(ex3)/lox(ex3)* | PMID:10545105 | RRID: MGI:2673882 | |
| Strain, strain background (*Mus musculus, 129S2/SvPasCrl*) | *Igf1r ⁺/⁻* | PMID:12483226 | RRID: MGI:3775301 | |
| Strain, strain background (*Mus musculus, C57/Bl6*) | TCF/LEF:H2B-GFP | PMID: 21176145 | Strain #:013752; RRID:IMSR_JAX:013752 | |
| Chemical compound, drug | Hoechst 33342 | Invitrogen | H3570 | 1:1000 |
| Other | Adeno-Associated Virus (AAV8-Cre) | Gene Transfer and Cell Therapy Core Facility, Faculty of Medicine, University of Helsinki | | 1:100 (stock: 1.13×10⁹ vg/µl) |
| Software, algorithm | Imaris | Bitplane | RRID:SCR_007370 | |
| Software, algorithm | Fiji | http://fiji.sc | RRID:SCR_002285 | |
| Software, algorithm | AfterQc | PMID:28361673 | RRID:SCR_016390 | |
| Software, algorithm | SortMeRNA | PMID:23071270 | RRID:SCR_014402 | |
| Software, algorithm | DEseq2 | PMID:25516281 | RRID:SCR_015687 | |
| Software, algorithm | Limma | PMID:25605792 | RRID:SCR_010943 | |
| Software, algorithm | biomaRt | PMID:16082012; 19617889 | RRID:SCR_019214 | |
| Software, algorithm | Salmon | PMID:28263959 | RRID:SCR_017036 | |
| Software, algorithm | Mfuzz | PMID:28263959;16078370 | RRID:SCR_000523 | |
| Software, algorithm | GraphPad Prism | GraphPad Software | RRID:SCR_002798 | |
| Software, algorithm | R Project for Statistical Computing | http://www.r-project.org/ | RRID:SCR_001905 | |

## Mice

To obtain mice constitutively expressing the Fucci2a cell cycle reporters (*Rosa26^Fucci2a del/Fucci2a del*), the conditional *Rosa26^Fucci2a flox/Fucci2a flox* mice (*Mort et al., 2014*) were first bred with Pgk1-cre mice (*Lallemand et al., 1998*) ubiquitously expressing Cre. The obtained Pgk1-cre;*Rosa26^Fucci2a del/Fucci2a flox* offspring were used to generate *Fucci2a* (*Rosa26^Fucci2a del/Fucci2a del*) mice without the Pgk1-cre transgene. Heterozygous *Rosa26^Fucci2a del/+* embryos were used for the quantitative analysis. The dual fluorescent *mGFP;mTmG* (*Rosa26^mGFP/mTmG*) mice were generated by breeding mTmG (*Rosa26^mTmG/mTmG*) mice (ICR background; the Jackson Laboratory Stock no. 007576) with *mGFP* (*Rosa26^mGFP/+*) mice (mixed background). The mGFP allele was generated by breeding mTmG mice with Pgk1-cre mice (*Lallemand et al., 1998*) to remove the sequence containing the mTdtomato coding region and STOP cassette surrounded by loxP sites leading to ubiquitous expression of mGFP. The obtained Pgk1-cre;*mGFP* mouse was bred with wild type C57Bl/6 mouse to remove the Pgk1-cre transgene. For embryonic tissue recombination experiments, male *mGFP;mTmG* mice were mated with wild type NMRI females. Krt14-Eda, where Eda is overexpressed under the control of Krt14 promoter in the developing ectoderm (*Mustonen et al., 2003*) and *Eda⁻/⁻* (The Jackson Laboratory, Strain #:000314) mice were maintained as described previously (*Voutilainen et al., 2012*). The Krt14-Eda;*Rosa26^Fucci2a del/+* embryos were

obtained by crossing Krt14-Eda males with $Rosa26^{Fucci2a\ del/Fucci2a\ del}$ females. As the *Eda* gene is localized in the X-chromosome, to obtain the $Rosa26^{Fucci2a\ del/+};Eda^{-/-}$ and $Rosa26^{Fucci2a\ del/+};Eda^{+/+}$ embryos, the $Rosa26^{Fucci2a\ del/Fucci2a\ del}$ mice were first bred with $Eda^{-/y}$ male or $Eda^{-/-}$ female to obtain $Rosa26^{Fucci2a\ del/+};Eda^{+/y}$ and $Rosa26^{Fucci2a\ del/+};Eda^{-/y}$ males, and $Rosa26^{Fucci2a\ del/+};Eda^{+/-}$ females. For the analysis, the $Rosa26^{Fucci2a\ del/+};Eda^{-/-}$ embryos were obtained by breeding $Rosa26^{Fucci2a\ del/+};Eda^{-/y}$ males with $Rosa26^{Fucci2a\ del/+};Eda^{+/-}$ females and $Rosa26^{Fucci2a\ del/+};Eda^{+/+}$ embryos were obtained by breeding $Rosa26^{Fucci2a\ del/+};Eda^{+/y}$ males with $Rosa26^{Fucci2a\ del/+};Eda^{+/-}$ females. The $Ctnnb1^{lox(ex3)/lox(ex3)}$ mice (*Harada et al., 1999*) were maintained in C57Bl/6 background as described previously (*Närhi et al., 2012*). $Ctnnb1^{lox(ex3)/lox(ex3)}$ or $Ctnnb1^{+/+}$ (wild type C57Bl/6) male mice were bred with C57Bl/6 wild type females to obtain the $Ctnnb1^{lox(ex3)/+}$ or $Ctnnb1^{+/+}$ embryos for the AAV virus transduction experiments. $Igf1r^{+/-}$ mice were maintained in 129S2/SvPasCrl background as described previously (*Holzenberger et al., 2003*). The littermates obtained from breeding of $Igf1r^{+/-}$ male and $Igf1r^{+/-}$ female mice were used for analysis. The *TCF/LEF:H2B-GFP* Wnt reporter mice (*Ferrer-Vaquer et al., 2010*) obtained from the Jackson laboratories (stock no. 013752) were maintained in the C57Bl/6 background.

All mice were kept in 12 hr light-dark cycles with food and water given ad libitum. The appearance of the vaginal plug was considered as embryonic day 0.5, and the age of the embryos was further verified based on the limb and craniofacial morphology and other external criteria (*Martin, 1990*). For embryos older than E13.5, only female embryos were used for experiments and analysis. The gender was determined by the morphology of the gonad as described previously (*Lan et al., 2022*) and further confirmed by detecting the Y chromosomal *Sry* gene using PCR (*Settin et al., 2008*).

## Ex vivo embryonic tissue culture and tissue recombination

Ex vivo culture of embryonic mammary glands was performed as described earlier (*Lan et al., 2022*). Briefly, the abdominal-thoracic skin containing mammary glands 1–3 was dissected from E13.5 to E16.5 embryos. The tissues were treated for 30–60 min with 2.5 U/ml of Dispase II (4942078001; Sigma Aldrich) in PBS at +4C in the shaker and then 3–4 min with a pancreatin-trypsin (2.5 mg/ml pancreatin [P3292; Sigma Aldrich] and 22.5 mg/ml trypsin dissolved in Thyrode's solution pH 7.4) at room temperature. The tissues were incubated in culture media (10% FBS in 1:1 DMEM/F12 supplemented with 100 µg/ml ascorbic acid, 10 U/ml penicillin and 10 mg/ml streptomycin) on ice for a minimum of 30 min before further processing. The skin epithelium was removed with 26 gauge needles leaving the mesenchymal tissue with the mammary buds.

For typical mammary gland culture, the tissues were collected on small pieces of Nuclepore polycarbonate filter with 0.1 µm pore size (WHA110605, Whatman) and further cultured on the air-liquid interface on filters with the support of metal grids in a 3.5 cm plastic Petri dish with culture medium. The explants were cultured in a humidified incubator at 37 °C with an atmosphere of 5% $CO_2$ and the culture medium was replaced every other day.

To test the role of IGF-1 in branching morphogenesis, explants were randomly separated into two groups. Mouse IGF-1 protein (791 MG, R&D systems) at the final concentration of 150 ng/ml was added to the culture medium 3 hr after the onset of the culture. The same volume of 10% BSA was used as a vehicle control. The fresh culture medium with IGF-1 or BSA was replaced after two days, and the explants were cultured for three days in total.

For tissue recombination experiments, embryos expressing mGFP or mTmG were identified with a fluorescent stereomicroscope and processed separately. Samples from $Igf1r^{+/+}$, $Igf1r^{+/-}$ or $Igf1r^{-/-}$ embryos were processed individually, and genotypes were verified by PCR before final analysis. The E13.5 submandibular glands (hereafter salivary gland) were dissected and processed similarly as described above for the mammary gland. After enzyme treatment and incubation on ice, the tissues were further dissected under a stereomicroscope to separate the intact mammary or salivary gland epithelium and their mesenchyme. The mesenchymes without any epithelium were collected with the filter and maintained in the culture incubator until further use. For salivary mesenchyme, mesenchymes from 3 to 4 salivary glands were pooled into one piece of filter to increase the amount of mesenchyme in each sample. After epithelial-mesenchymal separation of all samples, salivary epithelium or mammary buds 1–3 were gently washed by pipetting through a 1000 µl tip several times to remove the remaining mesenchymal tissues and then transferred onto the mesenchyme expressing different fluorescent protein, as previously described (*Lan and Mikkola, 2020*). 1–2 mammary buds were transferred to each mesenchyme. The recombinants were cultured as described above.

To specifically activate the WNT/ß-catenin signaling in the mesenchyme, tissue recombination has been performed as described above, while the mesenchymes from E13.5 *Ctnnb1^{lox(ex3)/+}* or *Ctnnb1^{+/+}* embryos were recombined with mammary buds from *Ctnnb1^{+/+}* embryos. Two hours after culture, final concentration of $1.13 \times 10^7$ vg/µl AAV8-Cre (purchased from AAV Gene Transfer and Cell Therapy Core Facility, Faculty of Medicine, University of Helsinki) were added into the culture medium. The fresh culture medium without virus was replaced every other day, and the explants were cultured 6–7 days in total.

## Time-lapse imaging for recombinants

To monitor the growth of the recombinants, the explants were imaged with Zeiss Lumar microscope equipped with Apolumar S 1.2 x objective once per day. To assess the branching type of each event of the epithelium in salivary mesenchyme, multi-position, automated time-lapse imaging described previously (*Lan et al., 2022*) was used instead. Briefly, tissue recombination was performed as described above (Day 0). One to 2 days after the culture, explants with filter were transformed to 24 mm Transwell inserts with 0.4 µm polyester membrane (CLS3450, Costar) and cultured on 6-well plates allowing multi-position imaging (*Lindström et al., 2022*). From day 1 or 2 to day 4 of culture, explants were imaged with 3i Marianas widefield microscope equipped with 10 x/0.30 EC Plan-Neofluar Ph1 WD = 5.2 M27 at 37 °C with 6% CO2. The medium was changed right before the imaging and thereafter, every other day. Images were acquired with an LED light source (CoolLED pE2 with 490 nm/550 nm) every 2 hr.

## Mesenchyme-free mammary rudiment culture and time-lapse imaging

E13.5 to E16.5 mammary rudiments were cultured in 3D Matrigel as previously described (*Lan et al., 2022*). Briefly, after separation of the mammary tissue with mesenchyme, the intact mammary rudiments 1–3 were dissected under stereomicroscope as described above. The mammary rudiments collected from littermate embryos of same genotype were pooled together, except for *Eda^{-/-}* or *Eda^{+/+}*. Pooled mammary rudiments 1–3 from each *Eda^{-/-}* and *Eda^{+/+}* embryo were cultured separately as it is not possible to obtain *Eda^{-/-}* and *Eda^{+/+}* genotypes from the same litter. Intact mammary rudiments were transferred onto the bottom of 12-well plates with 10 µl of culture media. The medium was then replaced with a 20–30 µl drop of growth-factor reduced Matrigel (356231; Corning) using a chilled pipette tip. The MBs were dispersed to avoid any potential contact with each other or the bottom of the plate. The mixture was then incubated in the 37 °C culture incubator for 15–20 min until the matrix was solidified. The MBs were cultured in a humidified incubator at 37 °C with an atmosphere of 5% $CO_2$ in serum-free DMEM/F12 medium supplemented with 1 X ITS Liquid Media Supplement (I3146, Sigma Aldrich) and 2.5 nM hFGF2 (CF0291, Sigma Aldrich), 10 U/ml penicillin and 10,000 µg/ml streptomycin. The culture medium was replaced every other day and the growth of the MBs was monitored once per day by imaging with Zeiss Lumar microscope.

## Whole-mount immunofluorescence staining and imaging

For whole-mount immunofluorescence staining, dissected ventral skin containing mammary glands, cultured explants, or mammary epithelia cultured in Matrigel were fixed in 4% PFA at 4 °C overnight, washed three times in PBS and then three times in 1% PBST (1% TritonX-100 in PBS) at room temperature. Samples were blocked with blocking buffer containing 5% normal donkey serum, 0.5% BSA, and 10 µg/ml Hoechst 33342 (Molecular Probes/Invitrogen) in 1% PBST at 4 °C overnight. The samples were then incubated with primary antibodies diluted in blocking buffer for 1–2 days at 4 °C, washed three times with 0.3% PBST at room temperature before incubation with secondary antibodies diluted in 0.3% PBST with 0.5% BSA for 1–2 days at 4 °C. After washing three times with 0.3% PBST and three times with PBS, samples were post-fixed with 4% PFA for 10 min at room temperature. Finally, samples were washed twice with PBS before immersing into the fructose-glycerol based clearing solution described by *Dekkers et al., 2019* before imaging. For samples from older embryos, the blocking step was extended to 2 days followed by an extra microdissection procedure, where samples were dissected under fluorescence stereomicroscope to expose the mammary epithelium and remove surplus mesenchymal tissues. The samples were imaged with Leica TCS SP8 inverted laser scanning confocal microscope with HC PL APO 20 x/0.75 IMM CORR CS2 object. The images were acquired with z-stack of 0.11 µm intervals.

For E13.5 *Igf1r* embryos, the staining was performed with the whole embryos before imaging. The samples of *Igf1r* embryos or IGF1-treated explants were imaged with Lumar stereomicroscope.

The following antibodies were used in this study: rat anti-mouse CD326 (EpCAM, 552370, BD Pharmingen, 1:500), rabbit anti-mouse Krt14 (RB-9020-P, Thermo Fisher Scientific, 1:500), rabbit anti-cleaved Caspase-3 (9661, Cell Signaling Technology, 1:500), Alexa Fluor 488-conjugated Donkey anti-Rat secondary antibody (A21208, Invitrogen, 1:500) and Alexa Fluor 647-conjugated Donkey anti-Rat secondary antibody (A48272, Invitrogen, 1:500).

## Image analysis

For mammary gland volume quantification, the border of mammary epithelium and mesenchyme was outlined manually based on EpCAM expression and bud morphology, and the surface rendering and volume quantification were performed with Imaris software (version 9.2, 9.5 or 10.0, Bitplane). The mammary gland tip number was counted manually in 3D using Imaris. To further quantify the cell cycle dynamics of mammary epithelial cells, the mammary epithelium was masked using the rendered mammary gland surface in Imaris. Epithelial cells expressing nuclear mCherry (G1/G0) or nuclear mVenus (S/G2/M) were automatically detected using spot detection function with manual correction. The distance of each detected nucleus to the mammary epithelium surface was measured using the distance transformation function of Imaris. To measure the cell volume, the 3D confocal images of EpCAM stained mammary glands were pre-processed for denoising using Noise2Void PlugIn (*Krull et al., 2019*) for ImageJ (Fiji, version 1.53t; *Schindelin et al., 2012*) with the N2V train and predict module. The training was performed with 100 epochs, 200 steps per epoch, batch size per step of 64, patch shape of 64, and neighborhood radius was of 4 or 5 depending on the quality of the images. Cells were then segmented in 3D with Imaris using Cell detection module. Manual examination was performed on segmented cells, and any segmentations of poor quality or cell volumes below 100 μm³ or exceeding 1000 μm³ were excluded from analysis. The TCF/LEF:H2B-GFP Wnt reporter expressing cells were detected using spot detection function with manual correction and the mesenchymal areas surrounding epithelia were masked using 50 μm cutoff ($V_{50μm}$) after distance transformation using rendered epithelial surface ($V_{epithelium}$) in Imaris. The volume information was extracted from Imaris and the volume of surrounding mesenchyme ($V_{surrounding}$) was calculated using formular $V_{surrounding} = V_{50μm} - V_{epithelium}$. To determine the percentage of cleaved caspase-3 positive cells in 3D cultured mammary epithelia, the total cell number was assessed with CellProfiler in 3D (*Carpenter et al., 2006*; *Jones et al., 2009*; *Lamprecht et al., 2007*), using the probability map of nuclei staining obtained from pixel classification with Labkit (*Arzt et al., 2022*) as input. The number of apoptotic cells was assessed manually with Imaris. All the data were exported to be further analyzed using R version 4.2.1, a free software environment available at https://www.r-project.org/.

To quantify the mammary gland growth affected by the deficient of *Igf1r*, the epithelial area of the mammary glands and the number of ductal tips were acquired manually with ROI Manager within ImageJ. The time-lapse images were pre-processed with StackReg PlugIn (*Thévenaz et al., 1998*) for ImageJ using Rigid Body transformation for drift correction, and then the branching types were identified by carefully inspecting the images from adjacent time points. Some time-lapse images were pre-processed for denoising similarly to what is described above.

The plots were produced with R using packages tidyverse version 1.3.2 (*Wickham et al., 2019*), ggplot2 version 3.4.0 (*Wickham, 2016*), ggsignif version 0.6.4 (*Constantin and Patil, 2021*), ggpubr version 0.4.0 (*Kassambara, 2023*) and RcolorBrewer version 1.1–3 (*Neuwirth, 2022*).

## RNA sequencing and data analysis

To obtain the mesenchyme samples for RNA sequencing, salivary glands or flank skins with mammary rudiments 1–3 were dissected and followed by enzyme treatment as described above for ex vivo embryonic tissue culture. E13.5 salivary gland mesenchymes were obtained after removing the salivary gland epithelium. For E13.5 and E16.5 mammary gland mesenchymes, after removing the skin epithelium, the mammary epithelium and its surrounding mesenchyme were isolated together with small scissors followed by removal of the mammary epithelium using 26 gauge needles (303800, BD Microlance). The E16.5 fat pad precursor was microdissected from the explants after enzyme treatment. The E13.5 ventral skin mesenchymes further away from the mammary gland region were collected as E13.5 skin mesenchyme. The mesenchymes isolated from two to three embryos from the

same litter were pooled together as one sample. Altogether, five biology replicates for each sample were collected from three different litters of C57Bl/6JOlaHsd mice. Samples were lysed immediately after collection and stored in TRI Reagent (T9424, Sigma) at –80 °C. Total RNA was extracted using Direct-zol RNA Microprep kit (Zymo Research, Irvine, CA) with DNase treatment according to the manufacturer's instructions. RNA quality was assessed with 2100 Bioanalyzer (Agilent, Santa Clara, CA) using Agilent RNA 6000 Pico Kit or Agilent RNA 6000 Nano Kit (Agilent, Santa Clara, CA). RNA concentration was determined using Qubit RNA HS Assay Kit (Q32855, Thermo Fisher) with Qubit 4 Fluorometer (Thermo Fisher). cDNA libraries were prepared with Ovation SoLo RNA-Seq System (NuGen/Tecan Genomics) according to the manufacturer's instructions and sequenced with NextSeq 500 (Illumina, San Diego, CA) in the DNA Genomics and Sequencing core facility, Institute of Biotechnology, HiLIFE, University of Helsinki. Forty-five to 68 million reads per sample were obtained after three rounds of sequencing.

For RNA-Seq data analysis, all sequencing reads were processed for quality control, removal of low-quality reads, adaptor sequence and ribosomal RNA using fastqc version 0.11.8 (*Andrews, 2010*), multiqc version 1.9 (*Ewels et al., 2016*), Trimmomatic version 0.39 (*Bolger et al., 2014*) and SortMeRNA version 2.1 (*Kopylova et al., 2012*) accordingly. The filtered reads were mapped to the reference genome (mm10) using Salmon version 0.99.0 (*Patro et al., 2017*) resulting in 36.6–53.4 million mapped reads per sample. The GSVA analysis was performed with R package GSVA version 1.44.5 (*Hänzelmann et al., 2013*). The conversion of murine gene Ensembl IDs to human Entrez IDs was performed with the biomaRt package version 2.46.3 (*Durinck et al., 2005*; *Durinck et al., 2009*), using the reference mart https://dec2021.archive.ensembl.org. The significant differentially expressed signatures between different mesenchymes were assessed with lmFit and eBayes functions from R package limma version 3.52.4 (*Ritchie et al., 2015*), by comparing E13.5 MM, E16.5 MM, or E16.5 FP with E13.5 SM, respectively. The signature database was downloaded from https://www.gsea-msigdb.org/gsea/index.jsp (*Subramanian et al., 2005*) on February 12, 2023. The significantly enriched KEGG signaling pathways were pooled together for visualization. The data normalization and analysis of differentially expressed genes (DEGs) were performed using the R package DESeq2 version 3.15 (*Love et al., 2014*). DEGs were defined with the thresholds of average count number >50, adjusted p-value <0.05 and Log2(Fold Change) ≥ 0.58 in each pairwise comparison.

Gene Ontology enrichment analysis was performed with the DEGs using R package pathfindR version 1.6.4 (*Ulgen et al., 2019*). Only the GOBP terms with lowest adjusted p value less than 0.01 were considered as significant. Among the commonly significantly altered GOBP terms, the top 10 GOBP terms with lowest adjusted p-value in each comparison and totally 16 GO terms were plotted. Gene Ontology database was downloaded from MSigDB (*Subramanian et al., 2005*) using R package msigdbr version 7.5.1 (*Dolgalev, 2022*) on November 9, 2022.

The DEGs with an average count number >100 and upregulated more than twice (Log2(Fold change) ≥ 1) in each group of samples compared to all the other four groups of samples were identified as marker genes.

To detect the pattern of the gene expression among different mesenchymal tissues, DEGs encoding extracellular matrix protein or ligands in selected pairwise comparisons with an average count number >200 in each group were further analyzed using Mfuzz version 2.58.0 (*Futschik and Carlisle, 2005*). The average of the normalized count number of each group was used as input. In addition, the groups were converted to pseudotime for the analysis. The fuzzifier m was determined with the default function and returned a value of 2.113207. The number of clusters was optimized empirically and set as 9 for the final analysis. The curated database including ECM, Ligand or Receptor genes was combined from the databases of R package SingleCellSignalR version 1.2.0 (*Cabello-Aguilar and Colinge, 2022*), CellTalkDB version 1.0 (*Shao et al., 2021*) and curated GO terms downloaded from https://baderlab.org/CellCellInteractions (*Qiao et al., 2014*).

The plots were produced using R packages tidyverse version 1.3.2 (*Wickham et al., 2019*), ggplot2 version 3.4.0 (*Wickham, 2016*), circlize version 0.4.15 (*Gu et al., 2014*), RcolorBrewer version 1.1–3 (*Neuwirth, 2022*), pathfindR version 1.6.4 (*Ulgen et al., 2019*), ComplexHeatmap version 2.12.1 (*Gu et al., 2016*), venn version 1.11 (*Dusa, 2022*) and patchwork version 1.1.2 (*Pedersen, 2022*).

## Public RNA-Seq data analysis

The raw data from *Wang et al., 2021* (OEP001019) were downloaded from https://www.biosino.org/node/index. The sequence reads were processed similarly as described above. The log2 transformed normalized expression of selected genes were extracted to construct the heatmap shown in *Figure 6F*.

## Statistical analysis

All data were analyzed by Prism 9 (GraphPad Software), or R packages ggsignif version 0.6.4 (*Constantin and Patil, 2021*) and ggpubr version 0.4.0 (*Kassambara, 2023*). Statistical tests used are indicated in figure legends. p-values <0.05 were considered significant. Throughout the figure legends: *p<0.05, **p<0.01; ***p<0.001, ****p<0.0001.

## Acknowledgements

The authors wish to thank Dr. Jianpin Cheng, Dr. Alison Kuony and M.Sc. Aida Kaffash Hoshiar for the critical comments and suggestions on the manuscript, Ms. Raija Savolainen and Ms. Merja Mäkinen for excellent technical assistance, Dr. Maria Voutilainen, Dr. Satu-Marja Myllymäki and Dr. Ana-Marija Sulić for technical advice, past and present members of the Mikkola lab for insightful discussions. We also acknowledge Dr. Rishi Das Roy for the important discussion on RNA-Seq data analysis and CSC – IT Center for Science, Finland, for computational resources. AAVs were provided by AAV Gene Transfer and Cell Therapy Core Facility, Faculty of Medicine, University of Helsinki. Confocal and widefield microscope imaging and image analysis were performed at the Light Microscopy Unit, Institute of Biotechnology, supported by HiLIFE and Biocenter Finland. RNA sequencing was performed in the DNA Sequencing and Genomics Unit at the Institute of Biotechnology, HiLIFE, University of Helsinki. This work was carried out with the support of HiLIFE Laboratory Animal Centre Core Facility, University of Helsinki, Finland. This work was supported by the Academy of Finland project grant (318287 to MLM) and Center of Excellence Program (307421 to MLM and JJ), the Cancer Society of Finland (MLM), the Jane and Aatos Erkko Foundation (MLM), the Sigrid Jusélius Foundation (MLM), the HiLIFE Fellow Program (MLM), Oskar Öflund Foundation (QL), the Doctoral Programme in Integrative Life Science of the University of Helsinki (ET), the Doctoral Programme in Biomedicine (MC), the Finnish Cultural Foundation (JS and BK), and Ella and Georg Ehrnrooth Foundation (JS). The funders had no role in study design, data collection and interpretation, or the decision to submit the work for publication.

## Additional information

### Funding

| Funder | Grant reference number | Author |
|---|---|---|
| Research Council of Finland | 318287 | Marja L Mikkola |
| Suomen Kulttuurirahasto | | Jyoti Prabha Satta |
| Ella ja Georg Ehrnroothin Säätiö | | Jyoti Prabha Satta |
| Oskar Öflunds Stiftelse | | Qiang Lan |
| Research Council of Finland | 272280 | Marja L Mikkola |
| Research Council of Finland | 307421 | Marja L Mikkola |
| Cancer Society of Finland | | Marja L Mikkola |
| Jane ja Aatos Erkon Säätiö | | Marja L Mikkola |
| Sigrid Juséliuksen Säätiö | | Marja L Mikkola |

| Funder | Grant reference number | Author |
|--------|------------------------|--------|
| Helsinki Institute of Life Science, Helsingin Yliopisto | | Marja L Mikkola |
| University of Helsinki | Doctoral programme in Integrative Life Science | Ewelina Trela |
| University of Helsinki | Doctoral Programme in Biomedicine | Mona M Christensen |
| Sigrid Juséliuksen Säätiö | | Jukka Jernvall |

The funders had no role in study design, data collection and interpretation, or the decision to submit the work for publication.

## Author contributions

Qiang Lan, Conceptualization, Data curation, Software, Formal analysis, Validation, Investigation, Visualization, Methodology, Writing – original draft, Writing – review and editing; Ewelina Trela, Jyoti Prabha Satta, Beata Kaczyńska, Mona M Christensen, Investigation, Writing – review and editing; Riitta Lindström, Investigation, Methodology, Writing – review and editing; Martin Holzenberger, Jukka Jernvall, Resources, Writing – review and editing; Marja L Mikkola, Conceptualization, Resources, Supervision, Funding acquisition, Writing – original draft, Project administration, Writing – review and editing

## Author ORCIDs

Qiang Lan (iD) http://orcid.org/0000-0002-7765-6767
Ewelina Trela (iD) http://orcid.org/0000-0002-8374-6136
Riitta Lindström (iD) http://orcid.org/0000-0001-5177-0564
Jyoti Prabha Satta (iD) http://orcid.org/0000-0003-1920-5658
Beata Kaczyńska (iD) http://orcid.org/0000-0002-2530-1841
Mona M Christensen (iD) http://orcid.org/0000-0003-0655-8665
Martin Holzenberger (iD) http://orcid.org/0000-0003-4869-725X
Jukka Jernvall (iD) http://orcid.org/0000-0001-6575-8486
Marja L Mikkola (iD) https://orcid.org/0000-0002-9890-3835

## Ethics

All mouse experiments were approved by the Laboratory Animal Center at the University of Helsinki and the National Animal Experiment Board of Finland with the licenses number KEK19-019, KEK22-014 and ESAVI/2363/04.10.07/2017. Mice were euthanized with $CO_2$ followed by cervical dislocation.

## Decision letter and Author response

Decision letter https://doi.org/10.7554/eLife.93326.sa1
Author response https://doi.org/10.7554/eLife.93326.sa2

# Additional files

## Supplementary files

• Supplementary file 1. The list of identified marker genes for each mesenchyme and their normalized expression value in each sample.

• Supplementary file 2. The results of mFuzz analysis shown in *Figure 5F* and the normalized expression value of each gene in each sample.

• MDAR checklist

## Data availability

The raw and processed RNA-Seq data created in this study have been deposited in the GEO database under the access code GSE225821.

The following dataset was generated:

| Author(s) | Year | Dataset title | Dataset URL | Database and Identifier |
|---|---|---|---|---|
| Lan Q, Trela E, Mikkola ML | 2024 | Identification of the mesenchymal signals regulating embryonic mammary gland development | http://www.ncbi.nlm.nih.gov/geo/query/acc.cgi?acc=GSE225821 | NCBI Gene Expression Omnibus, GSE225821 |

The following previously published dataset was used:

| Author(s) | Year | Dataset title | Dataset URL | Database and Identifier |
|---|---|---|---|---|
| Wang J, Song W, Yang R, Chao L, Wu T, Dong XB, Zhou B, Guo X, Chen J, Liu Z, Qc Y, Li W, Fu J, Zeng YA | 2021 | Fibroblast relays Wnt signals from endothelial niche to mammary epithelium | https://www.biosino.org/node/project/detail/OEP001019 | The National Omics Data Encyclopedia, OEP001019 |

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
