## [Editor Report]

In this valuable study, the authors use classical embryonic tissue recombination and pharmacological manipulation of explants in conjunction with cutting edge 3D imaging of tissue derived from sophisticated reporter and knock-out mouse models, as well as transcriptomic analyses, to delineate and dissect regulatory pathways critical for early mammary development, specifically focusing on cell proliferation, and ductal branching. The conclusions are convincing and the findings will be of interest to the community of biologists interested in the cellular and molecular mechanisms of (early) mammary gland development, as well as to a broader community of developmental biologists studying branching morphogenesis in tissues such as lung, kidney and salivary gland.

---

## [Decision Letter]

[Editors' note: this paper was reviewed by Review Commons.]

---

## [Author Response]

We would like to thank all the reviewers for their valuable and insightful comments on our manuscript. We greatly appreciate the positive recognition of our work.

In response to the reviewers' comments, we have effectively resolved the majority of raised issues and incorporated a substantial amount of novel data and text changes in the revised manuscript, as detailed below. However, as elaborated below, we find that some of the proposed analyses will not provide substantial novel insights and/or exceed the current scope of the manuscript.

As requested by the *eLife* editorial office, we have also simplified the title and clarified the mouse genetic nomenclature. The key points of the revision are listed below; new data included after the preliminary revision are underlined.

Removed the contents related to lineage segregationPerformed live imaging of E16.5 mammary epithelia recombined with E16.5 mammary mesenchyme (Updated Figure 4B and C, and Figure 4-video 1)Included data on the purity of tissue recombination experiments (New Figure 1—figure supplement 1A and B, and Figure 4—figure supplement 1)Examined the volume of mammary gland from E13.5 Eda^-/-^ and wild type control embryos (New Figure 3—figure supplement 1C)Examined the volume of mammary epithelial cells from E13.5 and E15.5 5 Eda^-/-^ and wild type control embryos (New Figure 3—figure supplement 1D and E)Examined the apoptosis of 3D cultured mammary epithelial organoids (New Figure 1—figure supplement 1D and E)Included the Axin2 expression in different mesenchymes from RNAseq data (New Figure 5—figure supplement 1C)Imaged and quantified TCF-Lef:H2B-GFP Wnt reporter expression in different mesenchymes (New Figure 6A and B)Imaged and quantified the volume of mammary gland 2 in Igf1r^+/+^ and Igf1r^-/-^ embryos at E13.5, E16.5 and E17.5 (New Figure 7G-I)Performed tissue recombination experiments involving epithelial and mesenchymal tissues from E16.5 Igf1r^-/-^ and control (Igf1r^+/+^ or Igf1r^+/-^) mammary glands (New Figure 7J and K)Optimized the visualization of some figures (Figure 3B, C, I, J and N, Figure 7C, E, F, H and I, and Figure 7—figure supplement 1A-C)

In addition, we report having identified one error when compiling the data from the imaging analysis and genotype information for statistical analysis of E16.5 Eda^-/-^ and Eda^+/+^ Fucci2a cell cycle data due to the mislabeling of some of the images. Upon correction, we observed that contrary to the original manuscript, the proportion of S/G2/M/ cells was reduced in E16.5 Eda^-/-^ mammary glands compared to the control embryos. However, activation of the basal cell-biased proliferation remained similar as in controls and in this respect, the key message remains valid. Additionally, we have included more samples to this dataset. Figure 3K and L have been updated in the revised manuscript and the text has been modified accordingly.

We sincerely apologize for this mistake. We have conducted a thorough double-check of all other analyses, and no additional errors were identified.

In summary, we are confident that the revised manuscript has been significantly improved. We believe it now meets the standards for publication in *eLife*.

Point-to-point response to Reviewer’s comments:

Reviewer #1 (Evidence, reproducibility and clarity (Required)):

Summary:

This manuscript by Lan et al. addresses the still incompletely resolved question as to how branching morphogenesis of the embryonic mammary epithelium is regulated at the molecular and cellular level. Using (combinatorial) primary explant cultures of wildtype and genetically engineered mouse embryos, in which the authors have developed a unique expertise over many years, together with imaging and RNAseq analyses, they (i) show that the timing of epithelial branching is dictated by the biological age of the epithelium, but that an epithelial-mesenchymal interaction is required to bestow branching ability on the mammary epithelium somewhere between E13.5 and E16.5, (ii) seek to determine if and how lineage and cell proliferation affect branching, (iii) show that while salivary mesenchyme can promote growth (i.e. branching density) of the E16.5 mammary epithelium, the mode of branching (i.e. lateral branching vs tip-clefting) is an intrinsic property of the mammary epithelium, (iv) use transcriptomics to identify genes that are likely to control either mammary- or salivary gland specific growth and/or branching patterns, (v) hypothesize that low levels of WNT signaling in the mammary gland mesenchyme (due to relatively high expression of WNT signaling inhibitors) are responsible for mammary specific branching, (vi) show that hyperactivation of WNT/CTNNB1 signaling in the mesenchyme indeed induces hyperbranching, (vii) identify Eda and Igf1 as putative mediators and paracrine signaling factors that regulate branching of the mammary epithelium upon secretion from the mesenchyme downstream of WNT/CTNNB1 signaling and (viii) show that mammary gland branching is impaired in Igfr1 null embryos.

Major comments:Overall, this is a solid study that is well controlled and technically of high quality. The Materials and methods should allow follow up and replication by others and the transcriptomic data have been made available via NCBI GEO. I think the authors convincingly demonstrate points (i), (iii), (iv) and (vi) and (viii). I have some questions regarding (ii), (v) and (vii) and (viii) that I will pose below.

We thank the reviewer for the careful assessment and recognition of our work. In the subsequent sections, we have tried to address all the concerns raised by the reviewer.

Re: (ii): The authors try to study the link between basal cell fate and branching. They use position of the cells (which they describe clearly and which is a good choice), since they cannot use specific markers due to the fact that the basal and luminal linages have not yet segregated at this point.

This part of the manuscript is not the most straightforward to follow. The most obvious experiment would have been to focus on the location of the cells and their associated cell cycle profile – but the authors themselves have just recently published a pre-print (their REF #54, now also out in JCB) that is an in-depth study of the link between cell proliferation + cell motility and branching, but this only becomes apparent in the discussion. In that sense, Figure 2 of the current manuscript is less novel, although it is nice to see that it holds up in a slightly different analysis.

We thank the reviewer for acknowledging our recently published work, which is focusing on the active branching phase during late embryogenesis/around birth. In the current proliferation analysis, however, our focus was on a different aspect of embryonic mammary gland development: understanding the mechanism underlying the ability to acquire competence to branch, i.e. how the epithelium changes between late bud and sprout stages. Our data obtained from tissue recombination and 3D culture experiments suggest that heterotypic mesenchymes or mesenchyme-free 3D organoid culture conditions do not provide sufficient signals to support branching of mammary epithelia before E16.5. We have modified the manuscript to better emphasize this point.

Instead of focusing on the cell cycle markers, the authors turn to a K14-Eda mouse model – which shows precocious branching and a temporary reduction in K8 expression. They also analyze Eda-KO embryos. Quite frankly, I find the authors' reasoning difficult to follow here and I cannot deduce how these experiments really address the question at hand (i.e. how lineage and cell proliferation affect branching), so I hope they can rewrite this section of the paper to make the arguments more clear and easy to follow for the reader who, at this point, knows little about Eda. For example, the authors present the argument that K14-Eda mice show a transient reduction in K8 expression – but we don't know if that also really means a (temporary?) change in (future?) luminal cell fate. In fact, since Eda later also makes an appearance as a candidate factor to be secreted by the mesenchyme together with Igf1, I wonder if their K14-Eda data would not be better suited to underscore that point instead and if the authors should perhaps eliminate this section altogether and just refer to their prior work in REF #45. If the authors think the current data add something more, than they need to be more explicit about this (and then also introduce the link to REF #45 in the Results section).

We agree with all the reviewers in that this part of the manuscript was not mature enough and provided only indirect evidence on the potential link between lineage segregation and branching ability. This is an important question in the field that merits a study of its own and should be addressed with better tools than those available to us at present. As suggested by reviewers #1 and #3, we have omitted this part in the revised manuscript.

Re: (v): Do the authors have any WNT/CTNNB1 target genes that they can include in their transcriptomics analysis to show that the WNT/CTNNB1 signaling levels are indeed lower in the mammary mesenchyme? Axin2 comes to mind, but there are some other negative feedback targets that are often induced across tissues, e.g. Rnf43 and/or Znrf3 and/or Sp5?E.g. to include in FIg6E?

In the original manuscript (lines 339-342), we had performed the GSVA analysis comparing the KEGG database, and the significantly altered pathways comparing different mammary mesenchymes with salivary gland mesenchyme were pooled and displayed as heatmap in Supplementary Figure 4b (Figure 5—figure supplement 1B in revised manuscript). The WNT signaling pathway was lower in the mammary mesenchyme, especially at E16.5.

As suggested by the reviewer, we have analyzed Axin2, the most commonly used readout of WNT/CTNNB1 signaling activity in our RNA-seq data that we include as a new Figure 5— figure supplement 1C in the revised manuscript. Axin2 data indicate that Wnt/β-catenin signaling activity is lower in the E16.5 fat pad, where branching takes place, compared to younger stages of mammary gland and the salivary gland. For the full revision, we analyzed the expression of TCF/LEF1:H2B-GFP Wnt reporter in the mesenchymes of E13.5 salivary gland and E13.5 and E16.5 mammary glands, and quantified the proportion of GFP+ cells in the region adjacent to the epithelium in 3D (within 0-50 µm from epithelial-mesenchymal border) , based on whole-mount images. The new Wnt reporter data are in line with the Axin2 RNA-seq expression data (New Figure 6A and B) indicating lowest level of Wnt signaling activity in the E16.5 mammary gland mesenchyme.

Re: (vii) and (viii): The authors convincingly show the phenotype of the Igfr1 KO mice, but I hope the authors concur that an epithelial only Igfr1 KO (or alternatively a mesenchymal only Igf1 KO, or epithelial/mesenchymal recombination experiments with WT vs IGFR1 null or IGF1 null tissue, or experiments with small molecule inhibitors of IGF1/IGFR1 signaling) would have given more solid mechanistic evidence regarding the presumed paracrine effect of IGF1 signaling. I am not asking the authors to perform another mouse experiment or even generate or use these conditional strains, but if the authors agree, then I do think this would merit some attention in the Discussion section. See also my comments regarding Eda in point 1.

As shown in the current manuscript, Igf1 is expressed in the mammary and salivary gland mesenchyme. This finding is in line with E14 in situ expression data available in Genepaint (https://gp3.mpg.de/results/Igf1) showing that overall in embryonic tissues, Igf1 is mainly produced in mesenchymal tissues. Of note, in Genepaint, a clear signal can be detected in the salivary gland mesenchyme, not the epithelium. Published E16 and E18 datasets indicate low level of Igf1 expression in the mammary epithelium (https://wahl-labsalk.shinyapps.io/Mammary_snATAC/). Hence, we conclude that Igf1 is mainly produced by mesenchymal cells. Instead, Igf1r appears to be rather ubiquitously expressed. A previous study assessed BrdU incorporation in Igf1r^-/-^ mammary buds at E14.5, and reported a specific proliferation defect in the epithelium, while no difference was detected in the mesenchyme (Figure 9, Heckman et al., 2007; PMID:17662267). However, we cannot exclude the possibility of autocrine, mesenchymal Igf1/Igf1r signaling, which in turn could lead to upregulation of a paracrine factor to regulate epithelial growth.

We agree with the reviewer in that novel conditional mouse models are beyond the scope of the current study. However, we do not think that small molecule drugs could be used to block Igf1r activity in a tissue-specific manner neither.

As proposed in the revision plan, to further delineate the paracrine and/or autocrine role of Igf1/Igf1r pathway during mammary epithelial growth and branching, we performed tissue recombination experiments between Igf1r^-/-^ and control (Igf1r^+/+^ or Igf1r^+/-^) mammary epithelium and mesenchyme, as suggested by the reviewer. The results show that the control epithelium grows equally well when recombined with control or Igf1r^-/-^ mesenchyme, while Igf1r^-/-^ epithelium fails to grow even if recombined with control mesenchyme (New Figure 7J and K). We conclude that epithelial Igf1r deficiency is the primary cause of the Igf1r^-/-^ phenotype.

Minor comments:A few minor spelling/grammar errors, including a couple of "the"s missing first line of the abstract, and also preceding "Majority" in line 148.

We apologize for these slips. They have been corrected in the revised manuscript.

Line 517-518: please also include the details for the Eda mice.

We apologize for missing this important information in Materials and methods. We have included a short introduction of the K14-Eda mice (renamed as Krt14-Eda upon editorial office’s guidelines), a new reference for the original publication producing them, as well as the Jackson Laboratories strain number for Eda^-/-^ (a.k.a. Tabby) mice in the revised manuscript.

1f spelling error: separation

The spelling error has been corrected in the revised manuscript.

Referees cross-commentingHaving read all three review reports I think they are pretty much in agreement, with shared questions about the inclusion/meaning/discussion of the lineage specification data and also agreement about the overall technical solidity of the data and this approach.I gather that reviewer #2 asks for more controls than myself or reviewer #3 and while I think all of their points are valid, in principle, I don't think all of these are required. I should add that I am inclined to trust the authors on their ability to separate mesenchyme and epithelium as they have been developing and optimising this system over many years.

We are grateful to the reviewer for the reliance on the technical aspect of our experiments. We do routinely monitor tissue purity in the recombinants (for more details, see our response to reviewer #2). To demonstrate this, we have included new data in new Figure 1—figure supplement 1A and B and new Figure 4—figure supplement 1. We believe these additions will further enhance the validity of our findings and effectively address the concerns raised by reviewer 2.

Reviewer #1 (Significance (Required)):General assessment:This is a carefully executed study in which an impressive amount of (combinatorial) embryonic mammary tissue explant experiments are combined with quantitative imaging and transcriptomics analysis.The main limitations of the work lie in the fact that the investigation of a potential link between branching and the cell cycle is not entirely novel, as the authors themselves recently published an nice pre-print (now also out in JCB) describing similar analyses. In addition, the mechanistic link between WNT/CTNNB1 signaling in the mesenchyme and the paracrine signaling activities of the presumed downstream effectors EDA and IGF, while plausible, is not yet complete. The work also does not yet addresses what exactly the branching identity is that is bestowed upon the mammary epithelium between E13.5 and E16.5 and how this then becomes an intrinsic (epigenetic?) feature of the mammary gland.

We thank the reviewer for acknowledging our recently published work. However, we wish to emphasize that our prior study focused on the active phase of branching morphogenesis. In contrast, the current study addresses the question of how the initially quiescent mammary bud acquires the competence to branch. These are two different biological questions. It is also worth noting that this is a rather minor component of the current manuscript.

Advance:This work provides more insight into the embryonic branching of the mammary gland – a stage of mammary gland development that is still poorly understood and that is, in general, understudied. In part, the work confirms prior work in the literature (their REF #19) regarding mammary and salivary gland tissue recombination experiments. It supplements this with a more elaborate time series of heterochronic and heterologous epithelium/mesenchyme explant cultures, using genetically engineered (and fluorescently labeled) mouse tissues to allow better and quantitative imaging.The transcriptomic analysis of different mesenchyme populations is also informative and allows the researchers to propose a putative mechanism for why the mammary gland branches differently from the salivary gland.The advance is both technical and functional, as well as conceptual, with some advance in terms of mechanism.Audience: This works should appeal to mammary gland biologists interested in the molecular and cellular mechanisms of (early) mammary gland development, as well as to a broader community of developmental biologists studying branching morphogenesis in tissues such as lung, kidney and salivary gland.My expertise:WNT signaling and mammary gland biology, at the intersection of developmental, stem cell and cancer biology.

Reviewer #2 (Evidence, reproducibility and clarity (Required)):The mammary gland is a branched structure that consists of a bilayered epithelium embedded in a specialized mesenchyme. In mice, at 11,5 days of embryogenesis, the ectoderm thickens forming 5 pairs of peculiar structures called placodes. During the following days, the placodes will grow and invaginate into the surrounding mammary mesenchyme and they will finally start to branch by the end of embryogenesis (E16). It has been suggested that the bidirectional communication between the growing mammary gland and the surrounding mesenchyme plays a pivotal role in the determination of each step of mammary gland development (placode formation, mammary bud invagination, gland outgrowth, branching). The role of different signalling molecules has already been shown, particularly for the placode growth and mammary bud invagination. Nevertheless, the pathways regulating embryonic mammary gland branching are still incompletely understood.In this manuscript, Lan and colleagues aim to decipher the correlation between different stages of mammary gland development such as proliferation, lineage segregation and ductal branching. Furthermore, they want to define which stage of mammary development is intrinsically determined by the epithelium and which one requires the supportive guidance of the mesenchyme. Lastly, they aim to discover the key signal for the growth and branching of mammary epithelium.To these purposes, they used an ex vivo model of heterochronic epithelial-mesenchymal recombination. In particular, they micro-dissected the epithelium and/or the mesenchyme from murine mammary glands at different stages of embryonic development (i.e. at E13,5 for the quiescent phase or 16,5 for branching phase) and explanted them together in different combinations using fluorescent reporters. To assess the role of the mesenchyme they also cultured the epithelium in a mesenchyme free 3d structure. Through this model they demonstrated that the presence of the mesenchyme is necessary for the priming of mammary epithelium for branching, since only E16,5 epithelial cells were able to grow and branch in a mesenchyme free 3D experiment. Nevertheless, intrinsic properties of the epithelium are necessary for the timing of branching, since E16,5 mesenchyme was not able to accelerate the outgrowth of E13,5 epithelia.In order to determine which epithelial properties are important, the authors correlated the beginning of cell proliferation in the embryonic mammary gland to the beginning of the branching phase. They indeed used the Fucci2a mouse model to carefully characterise the timing of mammary cells proliferation at different stages of embryonic development, concluding that the great majority of proliferating cells reside in the inner part of the mammary bud until E14,5, while in the external part at later stages.Regarding the importance of cell proliferation, Lan and colleagues claim that the beginning of the branching phase is not its direct consequence, thanks to the use of the K14Cre- Eda mouse model, known to have anticipated mammary gland development. Using this and the Eda-/- models, the authors also sustain that the branching occurs independently of the lineage specification of the epithelium.The use of salivary mesenchyme instead the mammary one was able to increase the number of branching of E16,5 mammary epithelium. Nevertheless, this model demonstrated that the branching pattern (side branching vs tip bifurcation) is an intrinsic feature of the epithelium. Lan and colleagues also defined the transcriptomic profiles of the mammary and salivary mesenchymes at different stages. In particular, they observed an increased expression of negative regulators of Wnt pathway in the mammary mesenchyme compared to the salivary mesenchyme. Moreover, using a mouse model where B-catenin is stabilised, they observed increased tip production in the mammary gland epithelium. They also showed that IGF1 production is increased after Wnt pathway activation and they tested its function, both treating their ex vivo cultures with exogenous IGF1 and using Igf1r-/- mouse models.

Major commentsThe great majority of the results of the manuscript are based on an ex vivo model of heterochronic epithelial-mesenchymal recombination. Since the authors are studying the effect of the mesenchyme of different stages on the epithelium (and vice versa), the purity of the two compartments after the dissection is particularly important. Although they said that the purity is evaluated (line 112), it would be important to show a control staining in which they use known markers of the mesenchyme with no colocalization with the fluorescent reporter of the epithelium.

We agree with the reviewer that the purity of the separated tissues is very important for our conclusions. This is why we have used genetically labeled tissues in all recombination experiments: the epithelium and the mesenchyme were always isolated from embryos ubiquitously expressing GFP or tdTomato. We find this the most reliable way to assess the origin and purity of the isolated tissues. If there was any carry-over mesenchyme isolated with the GFP+ epithelium, this would be revealed as GFP+ mesenchymal cells in the recombinants consisting of otherwise tdTomato+ mesenchyme. And vice versa: any carryover tdTomato+ epithelium isolated with the mesenchyme would be revealed as tdTomato+ epithelial cells in the recombinants. We apologize for not making this clear enough in the original manuscript. In the revised manuscript, we now provide confocal high-resolution images of the recombinants (new Figure 1—figure supplement 1A and B). The explants have been co-stained with the epithelial marker EpCAM, revealing a robust colocalization between the ubiquitously expressed florescent labels in the designated epithelial tissues and the EpCAM.

Another important point for understanding the quality and impact of these findings is to assess the similarities and differences, if there are, between the in vivo mesenchyme and the ex vivo one. Indeed, once explanted and put in culture, mesenchymal cells could change their transcriptomic profile and consequently change their signals to the epithelium. The authors should assess the expression of the genes and pathways studied during embryonic development in vivo.

The reviewer is correct in that the transcriptomes will likely undergo some changes when organs are cultured ex vivo. This is why RNA-seq was done on freshly isolated tissues.

Regarding the potential changes taking place ex vivo, however, we do not consider them relevant with respect to the questions we are addressing in this study. The reason is (as reported in the manuscript) that all control recombinations (homochronic recombinations such as E13 epithelium + E13 mesenchyme, E16 epithelium + E16 mesenchyme etc.) branched essentially as in vivo. Therefore, we find the results and conclusions made from the tissue recombination experiments solid.

The authors clearly showed that E16,5 epithelium is able to branch in a mesenchyme free 3D culture model, while epithelia from earlier stages don't. This led to the conclusion that mesenchyme is necessary for acquiring the branching ability. Nevertheless, the authors also said that early stages epithelia scarcely grow in the mesenchyme free 3D culture. Therefore, the lack of branching may be due to the lack of growth, if not the increase of death, of epithelial cells. The authors should quantify the size and the cell death of the epithelia in the different culture conditions and discuss better this point.

The reviewer is correct in that one of the key functions of the mammary mesenchyme up to E16.5 may be to provide survival signals for the epithelium, and this might explain why epithelia younger than E16.5 fail to grow/branch when recombined with salivary gland mesenchyme and in mesenchyme-free organoid culture.

To address this issue in the full revision, we assessed apoptosis in E14.5 and E16.5 mammary epithelia cultured in the mesenchyme-free 3D culture organoid set-up for 2 days. As shown in the New Figure 1—figure supplement 1D and E, a significant increase in cleaved-caspase 3+ cells was observed in E14.5 mammary epithelia compared to E16.5. However, 40% (7 out of 17) of the E14.5 samples exhibited very low levels of apoptosis, similar to the E16.5 samples, despite the absence of branching capacity. These data suggest that apoptosis may contribute to, but is unlikely to be the primary factor limiting the branching ability of E13.5E15.5 mammary epithelia in the mesenchyme-free 3D culture.

At the onset of the culture, the size of E16.5 mammary epithelia is larger than those isolated at E14.5, as shown in Figure 2B. Therefore, we do not think quantifying the size after the culture period would provide any additional insights.

The Fucci2a model allowed to assess the proliferation of embryonic mammary epithelium, showing that the great majority of proliferating cells are basal, at late stages of development (line 182). As it has already been shown, lineage specification is a late process during mammary gland development. The fact that the proliferating cells reside at the external part of the bud does not mean that they are basal cells yet. A p63/K8 staining could be important to understand if the increased proliferation occured in already specified basal cells or not.

Indeed, mammary lineage specification is a later process. As pointed out in the manuscript and by reviewer #1, the widely used basal and luminal lineage markers have not yet segregated to separate compartments at the developmental stages analyzed in our study, and therefore cannot be used as tools for this purpose. We would like to emphasize that in the manuscript, we analyzed the cells based on their position, and have used the term basal to indicate the basal position, not the prospective lineage. Accordingly, we used the term inner instead of luminal cells. We have further clarified this point in the revised manuscript.

The use of Fucci2a model showed that 20% of epithelial cells are proliferative at E13,5. This phase is considered as "quiescent" by the authors (line 120), but the moderate proliferation rate shown in this experiment demonstrated that it is not. A change of the nomenclature is needed.

We have removed the word “quiescent” from the text.

Through the use of K14-Eda and Eda-/- models, the authors claimed that the lineage specification is not a prerequisite for ductal branching. To support this point, they showed that the K14-Eda mice have an anticipated branching although the expression of K8 in the inner part of the bud is transitorily decreased. The authors link the K8 downregulation to a transient suppression of the luminal lineage, but this is clearly overclaimed. Although K8 is a known marker of luminal lineage, the downregulation of one marker is not sufficient to support their thesis. They should first check more markers and in particular critical regulators of luminal lineage as Notch1, Foxa1 and Elf5. Lately, the use of different models that drive embryonic epithelial cells to a forced lineage commitment (Notch1 or Δnp63 overexpression) would support more their claim.

As additional evidence, the authors showed that Eda is able to promote basal cell signature. Firstly, the authors should better explain why this point would support their thesis. Secondly, the supplementary figure 2b does not show which genes are taken into account to define the basal signature. A list of these genes would be helpful, as well as staining for some representative proteins.

We thank the reviewer for these constructive suggestions. We agree with all reviewers in that this part of the manuscript was not mature enough and provided only indirect evidence on the potential link between lineage segregation and branching ability. This is an important question in the field that merits a study of its own to be addressed with better tools than those available to us at present. As suggested by reviewers #1 and #3, we have omitted this part in the revised manuscript.

The authors used the same mouse models to assess the importance of proliferation in the determination of ductal branching and they claimed that proliferation is not a sufficient feature. This conclusion was supported by two observations. The first one is the fact that the K14-Eda model shows an increased cell proliferation at early stages compared to wt, coupled with anticipated branching. Secondly, although having smaller glands compared to wt and showing a delay in ductal branching, Eda-/- mice have an epithelial proliferation rate very similar to wt. Again, the conclusion that proliferation is not sufficient for branching is overclaimed. Firstly, the authors should explain how the buds in wt and Eda-/- mice have different sizes although the similar proliferation (increased cell death?, cellular volume?). Secondly, to support the thesis that proliferation is not sufficient for branching, functional experiments should be performed (see point 12). For instance, the short-time treatments with inhibitors or promotors of proliferation may help to understand the effective role of proliferation in the determination of branching.

We greatly appreciate the reviewer for raising this question. After careful re-examination of all data associated to this question, we identified an error that had occurred during compilation of the Fucci2a image analysis data and the related genotype information. This resulted in the mislabeling of some of the E16.5 Eda-/- and control samples. Upon correction, we observed that contrary to the original manuscript, the proportion of S/G2/M/ cells was reduced in E16.5 Eda-/- mammary glands compared to the control embryos. However, activation of the basal cell-biased proliferation remained similar as in controls and in this respect, the key message remains valid. Additionally, we have included more samples to this dataset. Figure 3K and L have been updated in the revised manuscript and the text has been modified accordingly.

We are very sorry for this mistake in the original manuscript. We have conducted a thorough double-check of all other analyses, and no additional errors were identified.

On the other hand, we are not claiming that proliferation is not important for branching, as obviously new cells are needed as building blocks of growing tissues. In a recently published paper, we have assessed the role of proliferation in branch point formation in embryonic mammary glands. Using mitomycin C to block proliferation, we showed that initiation of new branches occurs even when proliferation is blocked (Myllymäki et al., JCB2023, PMID:37367826).

The reviewer was also asking why Eda^-/-^ mammary primordia are smaller at E15.5-E16.5 despite similar proliferation rates. In the revised manuscript, we have quantified the volume of E13.5 Eda^-/-^ and control mammary buds and show that Eda^-/-^ buds are ~25% smaller (3.5 ± 0.8 x 10^5^ µm^3^ in Eda^-/-^ vs. 4.6 ± 0.7 x 10^5^ µm^3^ in control, mean ± SD) already at the bud stage (new Figure 3—figure supplement 1C and D).

We have also quantified the cellular size in Eda^-/-^ and control mammary glands at E13.5 and E15.5 (before onset of branching) and found that mammary epithelial cells in Eda^-/-^ embryos are ~15% smaller (new Figure 3—figure supplement 1E and F). Together, these data indicate that the smaller size of E15.5-E16.5 Eda-/- mammary glands is a combinatorial effect of the smaller mammary anlage at E13.5, smaller cell size, and reduced proliferation. These findings, while interesting on their own, do not challenge our conclusions regarding the link between onset of proliferation and acquisition of branching ability.

The heterotypic epithelial-mesenchymal recombination using the salivary gland is interesting. Nevertheless, some stainings to assess the purity of their systems are again required (e.g., marker of salivary epithelium to verify the purity of the mesenchyme and vice versa).

As mentioned above, all tissue recombination experiments were performed so that the epithelium and the mesenchyme originated from genetically labelled embryos expressing different fluorescent proteins. In the revised manuscript, we provide confocal images of the salivary-mammary tissue recombinants (new Figure 4—figure supplement 1), confirming the purity of the tissue compartments used in these experiments.

This model clearly showed that the mammary epithelium can form more branching when combined with the salivary mesenchyme. Moreover, the salivary epithelium preferentially branches through tip bifurcation, while mammary epithelium combined with the salivary mesenchyme has a mixed pattern of tip bifurcation and side branching (typical of the mammary gland). The authors thus concluded that the branching pattern is an intrinsic feature of the epithelium. However, a comparison between the percentage of tip bifurcation and side branching in the heterotypic combination and the homotypic combination between mammary epithelium and mammary mesenchyme is crucial to understand this point. Indeed, these results are not sufficient to exclude that the branching pattern is partially determined by intrinsic features and partially by extrinsic signals. The authors should carefully quantify the branching pattern in the homotypic combination and compare that to the heterotypic one. If the percentage of tip bifurcation do not change, their conclusion is correct; if this percentage increases in the heterotypic combination, it would be a sign of a partial effect of the signals of the mesenchyme.

We thank the reviewer for raising this question. We have independently generated data on the type of mammary gland branching events in two papers with somewhat different culture and imaging conditions (Lindström et al., BiorXiv 2022 and Myllymäki et al., JCB, 2023, PMID: 37367826). Both analyses showed that in embryonic mammary glands, the majority of branching events (~70%) occurs by side-branching. These data are in line with the current study that we have now complemented to include also the mammary-mammary recombination experiments (revised Figure 4-video 1 and revised Figure 4B).

Quantification of branching events revealed no significant difference in the type of branching events of mammary epithelia grown with salivary or mammary gland mesenchyme (revised Figure 4C), further supporting our initial conclusions.

Through the analysis of their transcriptomic data, Lan and colleagues found that the mammary mesenchyme expresses higher levels of negative regulators of Wnt pathway compared to the salivary mesenchyme. To demonstrate the value of their findings, they should confirm this in vivo, through staining of known Wnt proteins on the salivary and mammary mesenchymes at the embryonic stage.

In mammals, there are 19 Wnt ligands, over a dozen secreted Wnt inhibitors, 10 Frizzled receptors, two Lrp co-receptors, and numerous other pathway modifiers that contribute to the net Wnt signaling activity in a complex manner. Furthermore, it has been “notoriously difficult to generate useful antibodies to vertebrate Wnt proteins…In general, these sera do not detect endogenous Wnt proteins in cell extracts, nor do they detect Wnt proteins in tissues by staining techniques. Hence, there are few data on Wnt protein distribution in intact vertebrate animals.” This is a direct citation from the Wnt Homepage, maintained by the Nusse Lab; https://web.stanford.edu/group/nusselab/cgi-bin/wnt/reagents#antibod.

For all these reasons, we do not find this approach feasible nor informative.

Instead, in the revised manuscript, we report the expression levels of Axin2, the most commonly used transcriptional readout of canonical Wnt activity in our RNA-seq data (new Figure 5—figure supplement 1C). Axin2 levels are lowest in the E16 fat pad where mammary branching takes place, much lower than in any other tissues analyzed in the study. For the full revision, we have also analyzed the expression of TCF/LEF1:H2B-GFP Wnt reporter in the mesenchymes of E13.5 salivary gland and E13.5 and E16.5 mammary glands, and quantified the proportion of GFP+ cells in the region adjacent to the epithelium in 3D (within 0-50 µm from epithelial-mesenchymal border), based on whole-mount images. The new Wnt reporter data are in line with the Axin2 RNA-seq expression data (New Figure 6A and B) indicating lowest level of Wnt signaling activity in the E16.5 mammary gland mesenchyme.

Since the ability of the salivary mesenchyme to promote a higher rate of branching in the mammary epithelium, the authors wanted to assess what could be the role of Wnt signalling. To do so, they used a mouse model where B-catenin is stabilised, allowing an increased Wnt signalling in the mammary mesenchyme. As a result, they observed increased branching in the mammary epithelium. They also found that IGF1 is a ligand regulated by Wnt pathway in the mesenchyme. Therefore, the use of exogenous IGF1 in their ex vivo model was able to increase the branching of the mammary epithelium. Moreover, Igf1r-/- embryos showed a significant decrease of mammary gland branching. The conclusion based on these experiments was that the Wnt-Igf1-Igf1r axis plays a pivotal role in the promotion of mammary gland branching during embryogenesis. This conclusion is overclaimed for different reasons. Firstly, the normalization of the ductal branching to the body weight is insufficient to exclude that the impact of the Igf1r knockout may have severe consequences on the mammary gland formation, upstream of the ductal branching. Another parameter for this normalization is required (e.g., size of the bud before branching, proliferation status, etc).

We agree with the reviewer in that Igf1r knockout may affect mammary gland formation in multiple ways, and also prior to onset of branching, as already indicated in the original manuscript: “…smaller size of E14 mammary bud has been reported earlier…” (lines 375376) and ‘…mammary gland 3 that was consistently absent.’ (lines 391-392). To assess whether the reduced size and branching of E16.5/E18.5 Igf1r-/- mammary glands is merely a consequence of the smaller anlage, the revised manuscript includes new data reporting quantification of the volume of mammary gland 2 of Igf1r^-/-^ and wild type littermate embryos at E13.5, E16.5, and E18.5 from 3D confocal images of whole mount EpCAM stained mammary glands. As can be seen from the new Figure 7G and H, at E13.5, the mutant mammary buds are about 60% of the size of the controls, at E16.5, 25% and at E18.5 only 20 % revealing a progressive defect, indicative of a specific defect at the outgrowth and branching stage. This conclusion was validated by normalization to the body weight: at E13.5 the size of Igf1r^-/-^ mammary anlage did not differ from that of the wild type embryos (p = 0.11), at E16.5 the sprouts were smaller in the mutants, though the difference did not reach statistical significance (p = 0.08), while at E18.5, the Igf1r^-/-^ mammary glands were significantly smaller (p = 0.000021). We find these data compelling evidence for a specific role for Igf1r in outgrowth and branching of the embryonic mammary gland.

The use of alternative models to specifically knockout the receptor in the epithelium or the ligand in the mesenchyme (e.g. viruses) would be even more useful to specifically focus on the role of this pathway for ductal branching excluding side effects.

We thank the reviewer for this suggestion. Unfortunately, based on our experience, viral shRNA delivery is not sufficiently efficient for effective gene silencing, unlike Cre delivery for a gain-of-function approach (used in the current study to flox out exon 3 of β-catenin) in case where the endogenous pathway activity is very low and therefore, targeting even a subset of cells is sufficient for upregulation of paracrine factors.

As proposed in the revision plan, to further delineate the paracrine and/or autocrine role of Igf1/Igf1r pathway during mammary epithelial growth and branching, we performed tissue recombination experiments between Igf1r-/- and control (Igf1r^+/+^ or Igf1r^+/-^) mammary epithelium and mesenchyme, as suggested by the reviewer. The results show that the control epithelium grows equally well when recombined with control or Igf1r-/- mesenchyme, while Igf1r-/- epithelium fails to grow even if recombined with control mesenchyme (New Figure 7J and K). We conclude that epithelial Igf1r deficiency is the primary cause of the Igf1r-/- phenotype.

Another limit of this model is the fact that Igfr1 can be bound by Igf2 as well and we cannot exclude that this has an impact too (except if Igf2 is not expressed at this stage). A quantification of Igf2 expression may be useful.

Indeed, we cannot exclude the possibility that Igf2 could also play a role (Igf2 expression was similar to Igf1 in our RNA-seq dataset, see Figure 4—figure supplement 1), but the connection of mesenchymal Wnt signaling activity was to Igf1, not Igf2. In fact Igf2 was somewhat downregulated in Wnt3A treated sample reported by Wang et al. (Wang et al., 2021) (highlighted by an arrow in the revised Figure 6F). We have also clarified this point in the Discussion of the revised manuscript (lines 581-585 in the version where changes are indicated, or lines 538-542 in the ‘clean’ version of the revised manuscript).

From the experiments presented in this section it is clear that Wnt-Igf1-Igf1r axis has to be finely regulated to have the correct amount of ductal branching in the embryonic mammary epithelium. Nevertheless, the author just showed the RNA levels of Igf1 in the different compartments they have analysed. Stainings to see the effective presence of the ligand on the tissue is mandatory to clarify the role of this axis in the ductal branching in vivo.

Igf1-Igf1r signaling plays a critical growth promoting function during embryonic and postnatal development. The expression of Igf1 at RNA and protein level has been detected in almost all tissues in humans (Daughaday et al., Endocr. Rev., 1989; PMID: 2666112). Given that Igf1 is a secreted protein and multiple Igf binding proteins (Igfpbs) (that regulate the bioactivity of Igf1 by sequestering it) are expressed in the mammary and salivary gland mesenchyme (Figure 6—figure supplement 1), we find it unlikely that Igf1 staining would provide any additional information to the current study, as they cannot be used to assess the source of Igf1, nor the location of signaling activity.

Furthermore, as underlined by the authors, this axis is specifically important and upregulated in the salivary gland. Due the limit of the Igf1R-/- model, we cannot exclude that, although Wnt-Igf1-Igf1r axis is able to increase the branching ability of mammary epithelium, the normal branching rate observed in wt mice is due to other pathways.

We agree with the reviewer in that other pathways are also important in regulating normal mammary gland branching, for example, Eda/NF-κB and FGF pathways as we described in the Introduction. Our results do not exclude the possibility that also pathways other than Wnt regulate Igf1 expression. The reviewer is correct that if a paracrine factor is expressed in the salivary gland but not in the mammary mesenchyme, its physiological effect may be limited to the salivary gland. Indeed, cluster 5 identified by the mFuzzy analysis (Figure 5F) is likely to include some genes like that. This is why we decided to focus on cluster 6 genes like Igf1. In the revised manuscript, we have better highlighted the difference between cluster 5 and 6 genes.

Unfortunately, with the currently available tools, we cannot test the importance of the endogenous mesenchymal Wnt signaling activity by inactivating Wnt signaling activity specifically in the mesenchyme at the time point when branching begins. This would require an inducible mesenchymal Cre line (mesenchymal β-catenin is essential for the early fate specification of the primary mammary mesenchyme; Hiremath et al., 2012, PMID: 23034629), and conditional β-catenin null mouse. We do not have such mice available and we find that these experiments are beyond the scope of the current study.

Lastly, once claimed to have found the key factor necessary for ductal branching promotion, the authors should also test if the proliferation and lineage segregations are unaffected in this context, confirming their dispensable role claimed in the initial part of the manuscript.

Igf1/Igf1r is well-known for its growth promoting function via cell proliferation. We have no reasons to think that this would not be the case also in the mammary gland, and it was not our intention to give the impression that proliferation was not affected. In fact, Hiremath et al. (2012) already reported a defect in epithelial cell proliferation in Igf1r mammary buds at E14. Our key finding is that compared to other organs, the mammary gland is particularly sensitive to loss of Igf1r during branching morphogenesis. Finally, as pointed out earlier, better tools will be needed to assess the potential link between lineage segregation and onset of branching, a topic that we hope to address in the future.

Minor comments:An important paper on mammary gland ductal branching was published on Nature in 2017 by Scheele and colleagues and should be presented in the introduction, even though it is at later stages (after birth).

We thank the reviewer for the suggestion. In the revised manuscript, we have added the findings from Scheele et al. 2017 in the introduction.

In line 136 and 139 the authors referred to Figure 2 but it should be Figure 1

We apologize for these slips. They have been corrected in the revised manuscript.

The sentence on line 142 should be rephrased, since "advanced developmental stages" may be referred to pubertal development. The authors should specify that they are talking about embryonic development.

We apologize for the potential misunderstanding. In the revised manuscript, we have used the phrase “advanced embryonic developmental stage” to describe our conclusion more precisely.

Reviewer #2 (Significance (Required)):Overall, the authors concluded that embryonic mammary gland development and branching are extremely sensitive to the loss of IGF1, normally produced by the mesenchyme. The topic of the paper is interesting, the experimental approaches are well conceived, the data are convincing and the findings are of interest to developmental biologists. Nevertheless, there are some significant points that need to be further investigated before considering the manuscript suitable for publication:

We thank the reviewer for the careful assessment and positive feedback of our manuscript. We have already addressed most of the points raised and our revision plan includes additional points to be addressed.

Reviewer #3 (Evidence, reproducibility and clarity (Required)):Here the authors use classical embryonic tissue recombination and pharmacological manipulation of explants in conjunction with cutting edge 3D imaging of tissue derived from highly sophisticated reporter and knock-out mouse models and state of the art transcriptomic analysis to masterfully delineate and dissect regulatory pathways critical for embryonic mammary development. Specifically, they set out to parse regulation of proliferation from that of branch patterning.While it has long been established that epithelial-mesenchymal interaction is necessary for mammary branching this work shows by heterochronic recombination that initiation mammary branching is not advanced by mesenchymal stage. By examining Fucci2a embryos the authors demonstrate that branching is preceded by a significant increase in basal cellbiased proliferation but, through further analysis of Eda gain and loss of function mice, conclude that proliferation per se does not cause branching. They show by heterotypic recombination with salivary tissue that early mammary epithelia rudiments require their own mesenchyme for survival and that although later E16.5 rudiments expand more robustly when in contact with salivary mesenchyme they nevertheless retain their characteristic mammary branch pattern. Thus, they establish that initiation and patterning are intrinsic properties of the epithelium but that early survival and later expansion/proliferation is regulated by the mesenchymal context. By transcriptomic comparison of mammary and salivary mesenchyme they reveal that genes encoding canonical Wnt attenuators and antagonists are highly expressed in early mammary mesenchyme and drop as branching ensues. The low expression of these negative regulators of Wnt signaling in salivary mesenchyme is proposed as an explanation for its growth and branch stimulating capability. In keeping with these observations, the authors show that experimental activation of mammary mesenchymal Wnt signaling augments both growth and branching. Lastly, they identify transcriptomic changes in IGF1 coincident with the initiation of mammary branching and confirm its role by extending analyses of the effects of gain and loss of function of IGF1 on embryonic mammary development.This is a thorough, well-constructed paper that adds new knowledge and important conceptual nuance and mechanistic insight to classical findings on branch patterning. This work is a technical tour de force and backed by solid quantitative and statistical analysis throughout. Their experimental approach is superb and the conclusions are sound. Their findings will be of great interest to the community of mammary gland biologists and to the wider field of embryologists focused on early development of a broad range of ectodermal appendages.I have some minor criticisms that I believe can be quickly remedied in a minor rewrite and suggestions for the authors consideration to improve the manuscript discussion as follows:Minor issuesAbstract, line 37: The authors misuse the word "decompose" – it should be "deconstruct"

We thank the reviewer for pointing out our mistake, which we have corrected in the revised manuscript.

Results, p7 line 48: Add "The" to the sentence: "The majority…."

Corrected it in the revised manuscript.

P8 line 173 This sentence refers to Figure 2G which is a quantitative plot. I would suggest replacing the word "cluster" which implies a spatial organization with the word "subset" or "significant fraction" The spatial data in Figure 2d support basal bias but do NOT to my eye show any clustering – in fact the proliferative basal cells appear to be evenly dispersed within the basal layer.

We thank the reviewer for highlighting this aspect. We agree that “significant fraction” is a more suitable term than “cluster”.

P9 line 188: The statement on basal cell lineage specification needs a reference.

Following the suggestions from reviewers #1 and 3, we have removed the content about lineage segregation in Results, together with this sentence.

P10 line 201-216 I found the section on lineage specification (Figure S2) weaker than the rest and a distraction from the main thrust of the paper making it difficult for the reader to focus. I suggest omitting this section and supplemental figures associated with it altogether.

We agree with all reviewers in that this part of the manuscript was not mature enough and provided only indirect evidence on the potential link between lineage segregation and branching ability. This is an important question in the field that merits a study of its own that should be addressed with better tools than those available to us at present. As suggested by reviewers #1 and #3, we have omitted this part in the revised manuscript.

P9 line 190: "displays precocious onset of branching" it is sufficient to say: displays precocious branching – the use of both "precocious" and "onset' is redundant.P10 line 229 Similarly, delete "the onset of branching was delayed" it is sufficient to say:branching was delayed.

Both sentences have been corrected it in the revised manuscript.

P11 line 243: Delete "on the regulation of the" and substitute the word "to" in the sentence: "Next, we shifted our focus on the regulation of the branching pattern, which is thought to be determined by mesenchymal cues."

We have corrected it in the revised manuscript.

P11 line 241 subtitle and Figure 4 title: The disparity in titles here is jarring for the reader: Results text subtitle: "Salivary gland mesenchyme is rich in growth-promoting cues, but does not alter the mode of branch point formation of the mammary epithelium". Figure 4 Title: "Mammary mesenchyme is indispensable for the branching ability of the mammary gland". I suggest to the authors divide the figure as well as the text to make the two points indicated by their disparate titles separately.

We thank reviewer for the suggestion to clarify the Results part of the manuscript. As suggested, we have split the data under two separate subtitles, but due to limitations in figure numbers, we prefer to report these data in one figure panel.

P12 line 279 From here on out the manuscript has a tendency to use the term "growth" ambiguously – in many instances it is unclear do they mean expansion, proliferation, increased branch number/ morphology?? Please try to clarify.

Our aim is to use the term growth to mean tissue growth (expansion). We hope that this is clearer in the revised manuscript.

P16 line 341 use word "prompted" instead of word "promoted"

We thank reviewer for spotting out the slip, which we have corrected in the revised manuscript.

P16 line 382: include word "embryonic" before "mammary development"

We have modified the text in the revised manuscript.

Discussion P18 line 416: Add the words "later stage (E16.5)" to the sentence: "Importantly, we demonstrate that salivary gland mesenchyme could only promote the growth of later stage (E16.5) mammary epithelium"

We thank reviewer for the suggestion. We have modified the text in the revised manuscript.

P19 line 437: Given the authors statement "Instead, cell motility is critical for branch point formation in the mammary gland" they should consider a brief sentence mentioning their transcriptomic findings on cadherin 11 and Tenascin.

We thank the reviewer for appreciation of our transcriptomic data. In the revised manuscript, we have added the following text in discussion: “Accordingly, we observed significantly increased expression of cell migration promoting genes such as Cdh11 (encoding Cadherin 11), and Tnc (encoding Tenascin C) (Andrews et al., 2012; Midwood et al., 2016) in E16.5 mesenchyme compared to E13.5 (Supplemental Table 2).” (lines 522-525 in the manuscript with changes indicated, or lines 479-482 in the ‘clean’ version of the manuscript)

P19 line 451: Similarly, given their statement "This observation suggests that mammary epithelium itself carries the instructions dictating the mode of branching" they could consider their transcriptomic data on Ltbp1 in "mammary specific" clusters 7,8,9 as a matrix molecule initially expressed by mammary mesenchyme but which becomes expressed by luminal epithelial cells at precisely the time they acquire lineage specification and intrinsic branching capability.

This is an excellent suggestion. We have added following text in discussion: “It is worth noting that certain mesenchymal factors, such as Ltbp1, began transitioning towards epithelium-specific expression around E16.5 (Chandramouli et al., 2013). Exploring the potential impact of these factors on the self-instructed branching capacity of the mammary epithelium could yield valuable insights.” (lines 545-548 in manuscript with changes indicated, or lines 502-505 in the ‘clean’ version of the manuscript).

P20 lines 462-470 The authors should address their theory of Wnt suppression in the mammary mesenchyme in the context, albeit conflictingly, of earlier studies showing expression of Wnt signaling reporters, in either epithelial or mesenchymal locations during early stages.

We thank reviewer for the suggestion. In the preliminary revised manuscript, we report Axin2 expression data as new Figure 5—figure supplement 1C. Axin2 expression data suggest that Wnt/β-catenin activity is lowest in the E16.5 fat pad (where branching takes place) compared to all other tissues analyzed in the study. For the full revision, we have also analyzed the expression of TCF/LEF1:H2B-GFP Wnt reporter in the mesenchymes of E13.5 salivary gland and E13.5 and E16.5 mammary glands, and quantified the proportion of GFP+ cells in the region adjacent to the epithelium in 3D (within 0-50 µm from epithelial-mesenchymal border), based on whole-mount images. The new Wnt reporter data are in line with the Axin2 RNA-seq expression data (New Figure 6A and B) indicating lowest level of Wnt signaling activity in the E16.5 mammary gland mesenchyme.

We have discussed these results in the discussion in revised manuscript (lines 558-569 in the manuscript with changes indicated, or lines 514-526 in the ‘clean’ version of the manuscript).

Reviewer #3 (Significance (Required)):Here the authors use classical embryonic tissue recombination and pharmacological manipulation of explants in conjunction with cutting edge 3D imaging of tissue derived from highly sophisticated reporter and knock-out mouse models and state of the art transcriptomic analysis to masterfully delineate and dissect regulatory pathways critical for embryonic mammary development. Specifically, they set out to parse regulation of proliferation from that of branch patterning.This is a thorough, well-constructed paper that adds new knowledge and important conceptual nuance and mechanistic insight to classical findings on branch patterning. This work is a technical tour de force and backed by solid quantitative and statistical analysis throughout. Their experimental approach is superb and the conclusions are sound. Their findings will be of great interest to the community of mammary gland biologists and to the wider field of embryologists focused on early development of a broad range of ectodermal appendages.

We much appreciate the positive evaluation of our manuscript. We have addressed all the feedback provided by the reviewer 3 in the revised manuscript.

Field of expertise: Embryonic and adult mammary development, Wnt signaling, cell adhesion